# EFFEKT: Efficient Federated Knowledge Transfer to Foundation Models

## Abstract

Recent data protection laws have accelerated the adoption of Federated Learning (FL) for privacy-preserving decentralized training. Nevertheless, increasing model sizes imposes substantial computational demands on client devices, limiting FL applicability in resource-constrained settings. We introduce a novel multi-domain federated learning framework in which lightweight client-side proxy models collaborate with a server-side Foundation Model (FM) to learn new concepts without sharing private data. Our approach, EFFEKT, enables efficient server-side training of domain-specific LoRA adapters while preserving feature-space alignment between the FM and proxy extractors via novel bi-directional cross-distillation strategies. Experiments on multiple real-world datasets and deployments on low-power edge devices demonstrate improvements over state-of-the-art baselines in most domains considered while maintaining lightweight computation at the client side.

## 1 Introduction

Federated Learning (FL) has emerged as a successful paradigm in machine learning, addressing the critical need for privacy-aware distributed training of models. This approach typically involves local optimization on client devices coupled with server-side aggregation, dissemination, and control (Shenaj et al., 2023b). The key advantage of FL lies in its ability to train models using locally available data on client devices, thereby preserving privacy by eliminating the need to transmit sensitive information to a central server. While FL has proven effective for training lightweight models on mobile devices, the recent advent of high-performing yet extremely complex transformer-based foundation models presents a significant challenge. These sophisticated models, due to their sheer size and computational requirements, are often too complex for deployment and training on client devices, potentially limiting the applicability of traditional FL approaches.

To illustrate the practical implications of this challenge, consider a scenario where a technology company aims to offer a personalized remote image recognition service exploiting a powerful foundation model at server side. The local implementations of this service could be tailored to specific domains (*e.g.*, plant or insect species recognition, or vehicle model identification) and adapted to individual user data (that could have strong privacy constraints), while the central server maintains a more comprehensive, multi-domain model. In this context, the ability to improve the global model using local, domain-specific data in a privacy-preserving manner becomes crucial. However, the complexity of state-of-the-art vision models often precludes their direct deployment on user devices, necessitating novel federated learning techniques.

A viable solution for this issue is to deploy a simple CNN model on client devices, with its feature space aligned to that of a larger, more complex model. This idea is exploited by FedPromo (Caligiuri et al., 2025), where the lightweight network served as a proxy for the local training of a task-specific decoder, which was then reattached to the main model. While this approach allows tackling the computational complexity issues, from a performance viewpoint, it is still hampered by the issues of the simple averaging of network weights used at the server side, especially when client models become too misaligned (McMahan et al., 2017).

In this paper, we introduce a novel server-side aggregation approach that allows for better performance by replacing standard weight averaging with a distillation-based scheme. Our method builds upon the concept of using a simple network with features aligned to the complex one. However, at each federated round, we

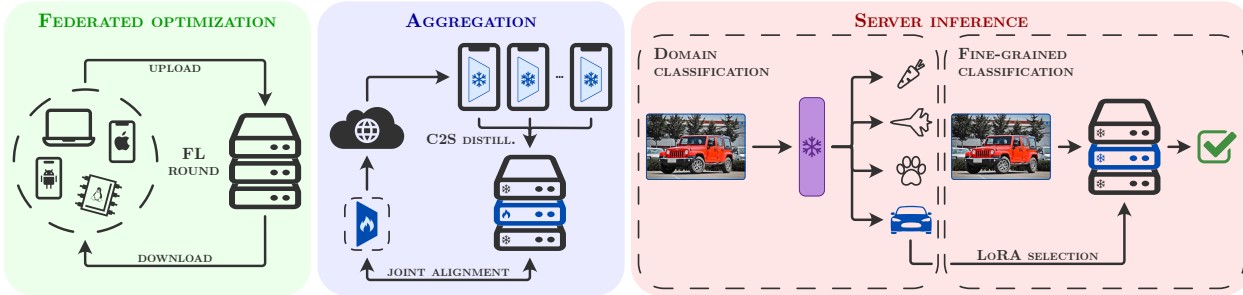

Figure 1: EFFEKT is a heterogeneous federated learning pipeline that allows bidirectional, multidomain optimization of a server-side foundation model and small client-side proxy models thanks to our Clients-to-Server (C2S) distillation and Joint Alignment (JA) objectives.

first efficiently distill the information learned by the clients into the foundation model, leveraging Low-Rank Approximation (LoRA). The updated foundation model is then utilized to perform joint re-alignment between the server and proxy models before initiating the new federated round. Note that this requires access to the latents and logits of the foundation model, which may not be available in API-based close-weight settings. Furthermore, access to task-specific pretraining domains remains a core feature of our approach, but, as discussed in Section A.5, the similarity requirement between pretraining and client data is not so stringent.

Crucially, our new approach yields substantial improvements in server-side accuracy, with an average increase over the state-of-the-art of 3.9% for top-1 accuracy and 2.7% for top-5 accuracy across 5 fine-grained domains. To further validate our setup and its real-world applicability, we deploy the architecture on compute-constrained devices. We measure final accuracy, energy consumption, and bandwidth usage in these deployments. Our results demonstrate that the simulated experiments closely mirror real-world deployments, exhibiting limited energy consumption and network usage, thus enabling deployment on personal devices.

This advancement not only enhances the performance of federated learning systems but also expands their potential applications. In the context of our earlier example, it allows for more effective learning of new, previously unknown classes at the server level without direct access to client data. This capability is particularly valuable in scenarios where the central model must evolve to recognize novel categories (*e.g.*, rare animal species or new vehicle models) based solely on distributed learning from user devices, all while maintaining strict privacy standards. Good privacy standards are guaranteed theoretically by the limited number of shared parameters and can be further improved at the price of a limited performance loss by adding the Local Differential Privacy technique (see Sec. A.4).

By bridging the gap between complex foundation models and resource-constrained edge devices, our approach paves the way for more inclusive and privacy-preserving AI applications, enabling a wider range of devices to participate in and benefit from advanced machine learning models.

## 2 Related Works

**Federated Learning (FL)** was introduced by McMahan et al. (2017), which proposed a simple server-side aggregation approach based on weight averaging. The approach, called FedAvg, was effective in simple scenarios but struggled when tackling strongly non-IID data distributions. The technique has been extended in many different directions (Yuan et al., 2024; Shenaj et al., 2023b). For example, FedProx Li et al. (2020) added a proximal term to the local objectives to limit the impact of local updates to reduce clients' drift. Similarly, SCAFFOLD (Karimireddy et al., 2020) tackles client drift through direct estimation and subsequent correction. MOON (Li et al., 2021) exploited model-based contrastive learning to improve client training.

**Foundation Models (FMs)** enabled significant improvements in several fields, from Natural Language Processing (*e.g.*, BERT (Devlin et al., 2019)) to Computer Vision. The CLIP (Radford et al., 2021) approach represents a key advancement in Vision-Language Models (VLMs); by using contrastive learning, it enables the alignment of visual and text representations. Another cornerstone work is DINO (Caron et al., 2021),

which learns the semantics of objects without supervision by utilizing self-distillation. FMs have also been used as oracle feature extractors for pretraining of other network models (Ostapenko et al., 2022).

**Pretraining in FL:** In the standard FL settings (McMahan et al., 2017), optimization begins with randomly initialized model weights; however, this can affect efficiency and performance, particularly in cases of high heterogeneity. To tackle this issue, server-side pretraining strategies have been employed to stabilize subsequent federated training, enabling longer local training and reducing communication costs (Shenaj et al., 2023b; Nguyen et al., 2023; Chen et al., 2023). Pretraining on synthetically generated data has also been utilized to establish a stable starting point for challenging vision applications (Shenaj et al., 2023a).

**Knowledge Distillation (KD):** Knowledge Distillation is a widely used strategy that facilitates the transfer of knowledge across different learning models (Hinton, 2015). It has recently been exploited in FL, both in a Model-agnostic way (Jeong et al., 2018; Li & Wang, 2019; Hongyan et al., 2019; Wu et al., 2024) to improve FL across various architectures and in a data-agnostic way (Zhu et al., 2021; Sattler et al., 2021; Zhang et al., 2022; Yao et al., 2023) to enhance performance in non-IID settings. Inspired by these works, our approach uses KD during pretraining to align the lightweight model with the server model.

**Federated Training of FMs** has recently emerged as a novel venue of research (Ren et al., 2025), focusing on aggregation techniques suitable for large-scale models and on improving both computational and communication efficiency. Some works concentrate on the federated training of specific FMs: a federated training strategy for CLIP (Li et al., 2021) is proposed in Lu et al. (2023); the work of Harasic et al. (2024) analyzes the performance of DINO (Oquab et al., 2023) in federated settings; a pretrained SAM (Kirillov et al., 2023) is fine-tuned on the clients in Liu et al. (2024). However, training these models is expensive; thus, other approaches tackle FM optimization by training LoRA adapters in a federated manner instead of the full models (Yi et al., 2023; Babakniya et al., 2023). Note that these methods train the adapter on the client and average the weights (*e.g.*, via FedAvg) on the server. Finally, Caligiuri et al. (2025) exploits a lightweight client architecture whose latent space aligns with that of a large model at the server, allowing federated training without bringing the FM to the clients. Iterating on these works, our approach trains a FM in a scalable, modular, and privacy-preserving manner, using FL on remote devices and KD at server side.

## 3   Problem Setting

In this section, we briefly recap the Cross-Architecture Federated Knowledge Transfer (CA-FKT) task (Caligiuri et al., 2025) and the mathematical notation.

Following traditional federated learning, we assume the presence of a server and a set of $n_k$ clients $\mathcal{K} = \{k_j\}_{j=1}^{n_k}$, which collaborate in a distributed optimization setup. Training is performed over $n_r$ communication rounds. In each round, the server selects a different set of $n_a$ active clients $\mathcal{K}_a$. Each client $k_j \in \mathcal{K}_a$ trains its model $M_{k_j} = H_{k_j} \circ E$ for $n_e$ local epochs using its private dataset $\mathcal{D}_{k_j}$ before sending the updated weights to the server for aggregation (note that only $H_{k_j}$ is trained). Then, the server samples a new client set $\mathcal{K}_a$ and sends to the clients the aggregated model as initialization for the next distributed optimization round. The setup introduces key differences from the standard approach.

Firstly, it supports domain-asynchronous training; therefore, the clients are divided into $n_i$ non-overlapping groups $\mathcal{K}^{d_i}$ based on their domain index $i \in \mathcal{I} = \{1, \ldots, n_i\}$. Since different domains require distinct sets of model parameters, for clarity of notation, we will add the domain index to the general notation introduced above for the rest of the document, *e.g.*, client $k_j$'s model $M_{k_j}$ becomes $M_{k_j}^{d_i} = H_{k_j}^{d_i} \circ E^{d_i}$, where $H_{k_j}^{d_i} : \mathbb{R}^{n_f} \mapsto \mathcal{Y}^{d_i} = \{1, \ldots, n_c^{d_i}\}$ and $E^{d_i} : \mathbb{R}^{h \times w \times 3} \mapsto \mathbb{R}^{n_f}$ are the domain $i$ head and encoder, respectively.

Secondly, we assume that the server and client model architectures are distinct; more specifically, the server encoder $\hat{O}$ is assumed to be much larger and better-performing than the small, efficient client encoders $E^{d_i}$. Crucially, to allow cross-distillation, the features produced by the two models must have the same dimensions, allowing us to attach the same head $H^{d_i}$ to both the server and client encoders. Refer to Sec. 4 for more details. While $\hat{O}$ is assumed to work well across all domains, to improve performance, we introduce a set of LoRA (Hu et al., 2022) adapters $\{L^{d_i}\}_{i \in \mathcal{I}}$. For ease of notation, we define $O^{d_i}$ as the server model adapted

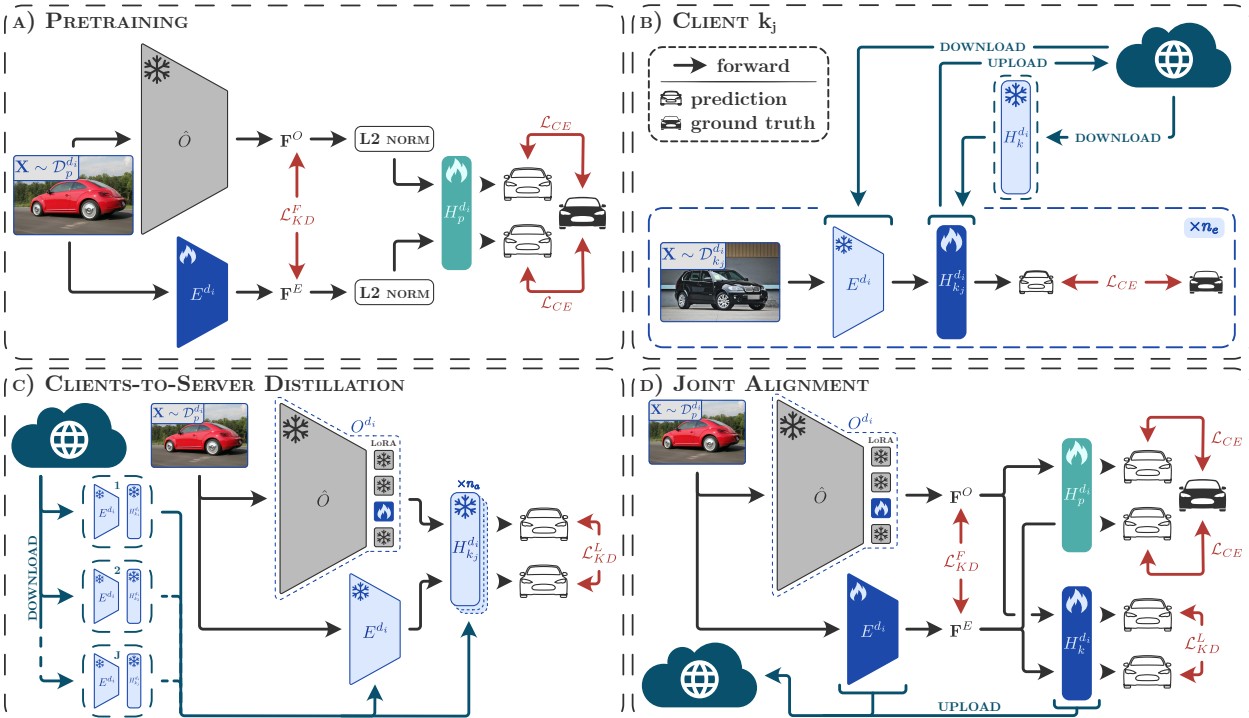

Figure 2: The EFFEKT architecture and cross-distillation strategies. a) pretraining pipeline; b) local update of client $k_j$. c) Clients-to-Server distillation (C2S), where the active clients' proxy models are used to update the correct domain-LoRA on the server; d) Joint Alignment (JA) distillation, utilizing the updated server model as initialization for the next round's proxy models.

with LoRA $L^{d_i}$. Formally, given an input RGB image $\mathbf{X} \in \mathcal{X} \subset \mathbb{R}^{h \times w \times 3}$ with a resolution of $w \times h$ pixels, the server and client encoders extract features $\mathbf{F}^O = O^{d_i}(\mathbf{X})$ and $\mathbf{F}^E = E^{d_i}(\mathbf{X})$, where $\mathbf{F}^O, \mathbf{F}^E \in \mathbb{R}^{n_f}$.

In our setup, during training, the domain-specific clients notify the server of the correct domain index $i$; while during inference, the domain of a sample $\mathbf{X}$ can be inferred using a domain discriminator $D : \mathbb{R}^{h \times w \times 3} \mapsto \mathcal{I}$ trained on a set of public, task-specific, pretraining datasets $\{\mathcal{D}_p^{d_i}\}_{i \in \mathcal{I}}$ with different, coarser, class labels.

## 4 Proposed Method

In order to tackle the CA-FKT task, we implement our EFFEKT federated framework (summarized in Fig. 2) on top of an encoder-decoder classification network, where the architectures of the server and client encoders differ while maintaining a common latent space. This choice permits the deployment of larger and higher-performing models on the server while maintaining a low computational profile on the clients. While, in principle, any couple of feature-aligned architectures could be used, to ensure fair comparison, we choose the same architectures used in competing works (Caligiuri et al., 2025). More specifically, the server architecture is a DINOv2 (ViT-L/14-Reg) model (Oquab et al., 2023), while the client model is a MobilenetV3-small (Howard et al., 2019) with a linear feature translator module. The content of the two feature spaces is aligned during a pretraining phase that is different for each domain $d_i$, and uses an appropriate public task-specific dataset $\mathcal{D}_p^{d_i}$. This results in a specific classification head $H_p^{d_i}$ for each of the domains due to the different class sets; the head can be freely attached to both server and client encoders.

### 4.1 Federated Training

Our implementation of the CA-FKT task aims to closely mimic a real-world deployment, which incurs several issues during distributed optimization, chief among them client divergence due to the strongly non-IID class

---

**Algorithm 1** Pseudocode for EFFEKT ServerRound.

---

**Require:** Number of rounds $n_r$ and active clients $n_a$, domain index $i$, server encoder (with LoRA) $O^{d_i}$, peak learning rate $lr_{\max}$, cosine annealing learning rate scheduler, pretraining head $H_p^{d_i}$

1: Select the domain clients' group $\mathcal{K}^{d_i}$, and the initial model $M_k^{d_i} = H_k^{d_i} \circ E^{d_i}$
2: **for** $r \leftarrow 1 \ldots n_r$ **do**
3:     Compute the active clients set $\mathcal{K}_a^{d_i}$ by randomly sampling $n_a$ clients from $\mathcal{K}^{d_i}$   // Different each round
4:     Send to the active clients $j \in \mathcal{K}_a^{d_i}$ the model $M_k^{d_i}$
5:     Compute current round learning rate from the scheduler $lr \leftarrow \text{CosineScheduler}(lr_{\max}, r, n_r)$
6:     **for** $k_j \in \mathcal{K}_a^{d_i}$ **do**                                    // Performed in Parallel
7:         Receive $H_{k_j}^{d_i} \leftarrow \text{ClientRound}\left(H_k^{d_i}, lr\right)$ from $k_j$
8:     **end for**
9:     $L^{d_i} \leftarrow \text{C2S}\left(L^{d_i}, E^{d_i}, \{H_{k_j}^{d_i}\}_{k_j \in \mathcal{K}_a^{d_i}}\right)$                          // Train LoRAs
10:     $H_k^{d_i} \leftarrow \text{FedAvg}\left(\{H_{k_j}^{d_i}\}_{k_j \in \mathcal{K}_a^{d_i}}\right)$                          // Aggregate heads
11:     $M_k^{d_i} \leftarrow H_k^{d_i} \circ E^{d_i}$                          // Update model with new head
12:     $\left(M_k^{d_i}, L^{d_i}, H_p^{d_i}\right) \leftarrow \text{JA}\left(M_k^{d_i}, L^{d_i}, H_p^{d_i}\right)$                          // Align
13: **end for**

---

distribution (Hsu et al., 2019). Therefore, strong regularization objectives and effective FL techniques are necessary to control the evolution of training. As such, we decided to employ a selective weight optimization strategy for the distributed optimization framework, *i.e.*, the ICP approach (Caligiuri et al., 2025). More details are in Alg. A.2 of the Appendix. Simply using this strategy, however, leads to subpar performance at the server side; therefore, we propose a novel cross-distillation federated aggregation technique to update the server model and improve its accuracy on unseen domains without impacting its generalizability using domain-specific LoRA adapters (Hu et al., 2022). More specifically, during each optimization round, each active client $k_j$ receives the updated model from the server and trains only the head $H_{k_j}^{d_i}$ using its private local dataset $\mathcal{D}_{k_j}^{d_i}$, thus preserving latent space alignment between different clients. After local optimization, the updated heads are sent to the server for aggregation and FM tuning. As detailed in Alg. 1, this step is split into two complementary operations: *Clients-to-Server (C2S)* and *Joint Alignment (JA)* distillation.

**Clients-to-Server (C2S) Distillation**
As the name implies, the first step of our aggregation pipeline involves updating the server-side FM by leveraging the knowledge acquired by the clients during the current round. Note that, to reduce computational and storage costs, the update is applied only to a small set of LoRA parameters. This choice is a good tradeoff between accuracy and complexity, as shown by the experiments in Tab. 7 of the ablation. The knowledge is updated using a logit-level knowledge distillation loss $\mathcal{L}_{KD}^L(\mathbf{P}, \mathbf{T}) = -\sum_{y \in \mathcal{Y}} \mathbf{P}[y] \log \mathbf{T}[y]$, where $\mathbf{P}$ is a vector of predicted class-probabilities and $\mathbf{T}$ is the reference distribution. Note that this is the reverse of the standard orientation, leading to several benefits in our setup. More specifically, this loss can be decomposed into $KL(P\|T) + H(P)$, that is, the mode-covering (Hinton, 2015) version of Kullback-Leibler divergence (useful under unreliable supervision) and prediction entropy, which is often used as a loss in unsupervised domain adaptation (Springenberg, 2015). This objective is computed on the output probability of each received (frozen) client head and is summed to compute the final loss:

$$\mathcal{L}_{C2S} = \sum_{k_j \in \mathcal{K}_a} \mathcal{L}_{KD}^L\left(H_{k_j}^{d_i}(\mathbf{F}_O), H_{k_j}^{d_i}(\mathbf{F}_E)\right) \quad . \tag{1}$$

Note that, since the training is performed by the server that does not have access to private client data, the samples used for distillation are extracted from the task-specific pretraining dataset $\mathcal{D}_p^{d_i}$. A more detailed breakdown of this procedure is reported in Alg. A.3 of the Appendix.

**Joint Alignment (JA) Distillation**
In general, the updates applied to the server model in the previous step may lead to misalignment between the feature spaces of the FM and of the client-side proxies. Therefore, an additional realignment procedure is

necessary to restore feature matching and improve the accuracy of the aggregated client model. Note that, in the previous step, the trained client heads were kept separate to preserve all available client knowledge. In contrast, here we begin by computing a single domain client head $H_k^{d_i}$ by aggregating the trained client heads $H_{k_j}^{d_i}, k_j \in \mathcal{K}_a^{d_i}$ using the FedAvg strategy (McMahan et al., 2017) to serve as the distillation target. This is necessary because a unique model must be sent to all active clients to initialize the subsequent federated round. As before, to preserve the privacy of client data, this realignment is performed by the server using the task-specific pretraining data $\mathcal{D}_p^{d_i}$. The key differences lie in the optimization targets, which include both classification accuracy and alignment objectives. We train the task-specific head $H_p^{d_i}$, the full client model $M_k^{d_i}$ (including the encoder), and the LoRA $L^{d_i}$. The training loss has two major focuses. The first being the classification accuracy of both the adapted FM and the client encoder using standard cross-entropy objectives. The second is the preservation of feature compatibility, which is achieved by encouraging server and client latent representations to be similar using the $\mathcal{L}_{KD}^F$ loss together with bidirectional logits alignment $\mathcal{L}_{KD}^L$. The overall loss function for this step can be written as:

$$
\begin{aligned}
\mathcal{L}_{JA} = \mathcal{L}_{CE}\left(H_p^{d_i}(\mathbf{F}_O), y\right) + \mathcal{L}_{CE}\left(H_p^{d_i}(\mathbf{F}_E), y\right) + \mathcal{L}_{KD}^F\left(\mathbf{F}_E, \mathbf{F}_O\right) + \mathcal{L}_{KD}^F\left(\mathbf{F}_O, \mathbf{F}_E\right) \\
+ \lambda_{KD}^L \mathcal{L}_{KD}^L\left(H_k^{d_i}(\mathbf{F}_O), H_k^{d_i}(\mathbf{F}_E)\right) + \lambda_{KD}^L \mathcal{L}_{KD}^L\left(H_k^{d_i}(\mathbf{F}_E), H_k^{d_i}(\mathbf{F}_O)\right) \quad ,
\end{aligned}
\tag{2}
$$

where $\mathcal{L}_{CE}$ denotes the standard categorical cross-entropy loss (applied to $H_p^{d_i}$ to match the class-set of the only domain dataset available at server-side, $\mathcal{D}_p^{d_i}$), $\lambda_{KD}^L$ is a hyperparameter, and $\mathcal{L}_{KD}^F$ is the sum of the L1, L2 and cosine distances (Barbato et al., 2024). The first two objectives are necessary to preserve the discriminative capability of the server and proxy encoders, respectively; the next two allow for shared application of the head; the last two update the aggregated head to ensure compatibility with the new latent space. The pseudocode is provided in Alg. A.4 of the Appendix.

## 4.2 Multi-Domain Setup

In our setup, each client knows its domain index and can communicate it to the server during training (this does not break privacy, as the domain categorization is not sensitive), thereby avoiding inter-domain confusion during aggregation. Note that in a realistic scenario, federated learning is often performed in a domain-asynchronous manner, meaning that the server must be able to activate the correct LoRA adapter and model head to optimize based on the received client updates. Our approach employs LoRAs and distinct heads instead of directly updating the server model, which guarantees the possibility of incrementally learning new domains without impacting the performance of previously learned models or compatibility with the compute-constrained proxy encoders.

**Multi-Domain Inference**
Whenever the server needs to run inference on a given sample, two scenarios are possible: either the prompt (from a client) includes the expected domain, or the domain information is missing. In the former case, since the client provided the necessary domain index, the server can directly activate the correct LoRA adapter and head before executing the model. In the latter case, the server must estimate the correct domain before proceeding. To tackle this issue, we introduce a server-side classifier that identifies the domain of the query.

**Domain Discriminator**
Since the domain discriminator may be utilized for each sample during server-side inference, we aimed to minimize its computational impact without compromising accuracy. Moreover, to avoid retraining the model every time a domain is added, we adopted a few-shot-based approach, *i.e.*, prototypical classification. More specifically, the domain classifier is implemented as a frozen encoder-only MobileNetV3-small (Howard et al., 2019) architecture pretrained on ImageNet-1k (Deng et al., 2009), which extracts a latent representation of each given query sample. At inference time, the extracted features are compared to a set of internally stored prototypes (one for each domain) using the L2-distance metric. The domain index $i$ corresponding to the closest prototype is selected as output. The domain prototypes are computed during the pretraining step of the domain-specific client encoders using the task-specific pretraining datasets $\mathcal{D}_p^{d_i}$. Despite the negligible computational cost, this simple classifier achieves remarkable domain accuracy, as detailed in Table 3 of the ablation studies.

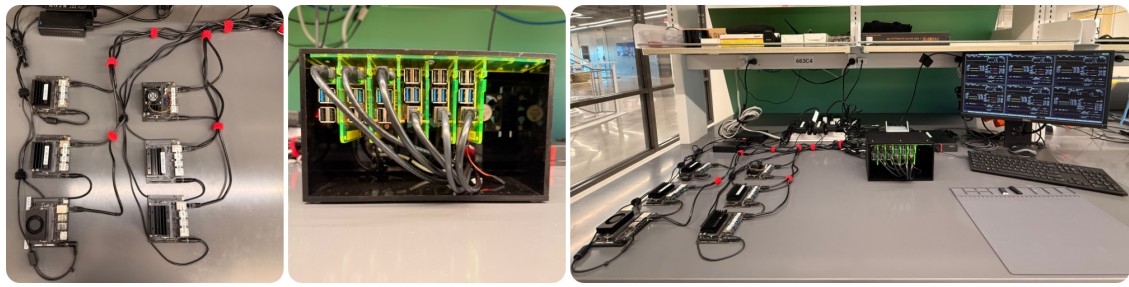

Figure 3: Real devices' setup: 3 Raspberry Pi 4, 3 Raspberry Pi 5, 5 Jetson Nano, and 1 Jetson Orin-Nano.

## 5 Implementation Details

We implemented our approach[1] using the *Flower* framework (Beutel et al., 2020) in *PyTorch* (PyTorch maintainers and contributors, 2016). Pretraining and simulated federated experiments utilized a single NVIDIA L40s GPU, running both the server and active clients in parallel. The server-side computational costs are detailed in Appendix A.6. Real-world experiments employed a server containing a single NVIDIA A100 GPU, and a cluster of 3 Raspberry Pi 4, 3 Raspberry Pi 5, 5 Jetson Nano and 1 Jetson Orin-Nano, all having a minimum of 4GB of RAM (the setup is shown in Figure 3). The supplementary material contains a short video showing the setup and training on real devices.

**Lightweight Encoders Pretraining**
We pretrain the lightweight encoders on the server for 60 epochs using a batch size of 64. We adopt a cosine-annealing learning rate scheduler (decaying to $lr = 0$) with a peak learning rate of $lr_{\max} = 0.005$. Optimization is performed using Adam ($\beta_1 = 0.9$, $\beta_2 = 0.999$) with no weight decay. We follow the TorchVision benchmark (TorchVision maintainers and contributors, 2016) for MobileNet data augmentation. As a reference, pretraining requires 40 minutes on an NVIDIA L40s for StanfordCars, while timing on other datasets scales roughly with size (see the Appendix).

**Simulated Federated Experiments**
Optimizer, scheduler, batch size, and peak learning rate are kept the same as in pretraining. The optimization runs for $n_r = 500$ rounds, with $n_e = 10$ local epochs per client (without augmentation). We consider a federated setup with $n_k = 100$ clients, where $n_a = 10$ clients participate at each round. The server-side distillation learning rate is set to $10^{-5}$ and $\lambda_{KD}^L = 0.01$. Federated finetuning on CompCars requires ~15 hours. The LoRA adapters are applied to each attention layer of the server model, targeting $Q$ and $V$. The rank is set to 16 for a total of $1.57M$ parameters ($\sim 194\times$ less than the full server model and $1.6\times$ less than the client model). See Table 7 for additional configurations.

**On-Device Federated Experiments**
The training configuration is analogous to that used in the simulated experiments, the key difference being that no GPU acceleration is employed on the clients. We set the rounds to $n_r = 300$, with $n_e = 10$ local epochs per client (without augmentation). We consider a federated setup with $n_k = 12$ clients, where $n_a = (2, 4)$ clients participate per-round. On-device federated finetuning on CompCars requires ~4 days. The peak learning rate was set to $lr_{\max} = 6 \times 10^{-4}$, and the batch size to 48. The number of active clients was chosen to preserve approximately the same ratio as the simulated experiments (1/10), and the learning rate was chosen to achieve, on average, the same gradient updates.

**Datasets**
To ensure fairness in the experimental evaluation, we evaluate EFFEKT on the same five domain pairs used by Caligiuri et al. (2025). All datasets exhibit long-tailed class distribution, having an average of 272 target classes belonging to similar objects, and are distributed across clients according to a Dirichlet distribution with concentration parameter $\alpha = 1$ (Hsu et al., 2019). In each scenario, a public source dataset is used for pretraining, while a private target dataset is distributed across clients in a federated setting. More specifically,

---

[1]Code will be released upon acceptance.

the dataset couples are organized as follows: CUB200 (Wah et al., 2011) → NABirds (Van Horn et al., 2015), ImageNetPets (Caligiuri et al., 2025) → OxfordPets (Parkhi et al., 2012), StanfordCars (Krause et al., 2013) → CompCars (Yang et al., 2015), FGVCAircraft (Maji et al., 2013) → MilitaryAircraft (Nakamura, 2024), and Food101 (Bossard et al., 2014) → UECFOOD256 (Kawano & Yanai, 2014).

## 6 Experimental Results

To evaluate the performance of EFFEKT, we compare it against six competing federated strategies. The first competitor is FedAvg (McMahan et al., 2017), a common federated learning baseline. The second, FedAvg + EMA, is an extension that tackles client divergence by applying Exponential Moving Average (EMA) to the model weights. The third (FedProx (Li et al., 2020)) and fourth (MOON (Li et al., 2021)) competitors are benchmark distributed optimization techniques that actively tackle client divergence by analyzing differences in the received weights. We consider the easiest (higher-accuracy) version of FedProx, where no *straggler* clients are present. As the fifth competitor, we selected the recent FedHEAL (Chen et al., 2024) approach, which extends the standard FedAVG aggregation, introducing a similarity-based re-weighting factor. The last competitor is FedPromo (Caligiuri et al., 2025), which introduced the ICP regularization we deploy at the client-side. Finally, we provide the performance of a centralized configuration as a reference. Note that all competitors are tested under the same conditions: *i.e.*, frozen proxy encoders pretrained with distillation on the public source dataset and classifier-only client training, while metrics are computed on the server by attaching the classifier to the Foundation Model. Table 1 compares the Top-1 and Top-5 classification accuracy of the server model $H_k^{d_i} \circ O^{d_i}$ with those of the competitors. Finally, we report the performance of a centralized experiment implementing the EFFEKT bi-directional distillation. Here, a single client trains its head on the full private dataset $\mathcal{D}^{d_i}$ for $n_r = 500$ rounds, while the server updates the domain-specific LoRAs.

### 6.1 Single Domain Results

Our approach surpasses competitors in most of the considered domains. In particular, in the

Table 1: Federated Top-1 and Top-5 accuracy (↑) on the server after federated aggregation. **Best** in bold, second-best underlined.

| Dataset $\mathcal{D}$ | Method | DINOv2 | |
| | | Top-1 | Top-5 |
|---|---|---|---|
| CompCars | FedAvg | 14.0 | 27.4 |
| | FedAvg + EMA | 21.5 | 42.6 |
| | FedProx | 13.7 | 31.2 |
| | MOON | 13.8 | 27.4 |
| | FedHEAL | 13.9 | 27.0 |
| | FedPromo | 35.8 | 69.4 |
| | EFFEKT (ours) | **43.0** | **75.0** |
| | Centralized | 43.2 | 73.6 |
| UECFOOD256 | FedAvg | 29.6 | 50.4 |
| | FedAvg + EMA | 49.1 | 83.1 |
| | FedProx | 8.2 | 42.4 |
| | MOON | 29.7 | 50.5 |
| | FedHEAL | 29.5 | 50.2 |
| | FedPromo | 58.7 | 88.7 |
| | EFFEKT (ours) | **62.1** | **88.8** |
| | Centralized | 62.8 | 89.3 |
| NABirds | FedAvg | 6.1 | 12.9 |
| | FedAvg + EMA | 20.7 | 43.6 |
| | FedProx | 16.0 | 39.9 |
| | MOON | 6.1 | 12.9 |
| | FedHEAL | 6.2 | 13.0 |
| | FedPromo | 30.3 | 65.0 |
| | EFFEKT (ours) | **39.1** | **72.8** |
| | Centralized | 37.9 | 69.2 |
| Military Aircraft | FedAvg | 9.2 | 27.7 |
| | FedAvg + EMA | 12.4 | **39.0** |
| | FedProx | 10.4 | 33.8 |
| | MOON | 9.6 | 27.8 |
| | FedHEAL | 9.5 | 27.9 |
| | FedPromo | 11.9 | 32.1 |
| | EFFEKT (ours) | **12.5** | 33.7 |
| | Centralized | 13.6 | 34.2 |
| OxfordPets | FedAvg | 47.9 | 79.2 |
| | FedAvg + EMA | 67.6 | 96.4 |
| | FedProx | 68.0 | 95.0 |
| | MOON | 47.4 | 78.8 |
| | FedHEAL | 50.1 | 79.5 |
| | FedPromo | **72.7** | **98.0** |
| | EFFEKT (ours) | 72.3 | 96.6 |
| | Centralized | 80.0 | 99.2 |

StanfordCars→CompCars scenario, EFFEKT's improvement over the second-best approach (FedPromo) reaches 7.2% and 5.6% Top-1 and Top-5 accuracy, respectively. A similar result is achieved in the Top-1 accuracy of the Food101→UECFOOD256 scenario, where the improvement is 3.4%. Top-5, instead, gains a more limited 0.1%. The highest gains are observed in the CUB200→NABirds scenario, where EFFEKT surpasses

Table 2: Accuracy on multi-domain scenario. *Known Domain*: query sample includes domain index $i$. *Unknown Domain: D* used to estimate $i$ (using pretraining in-domain dataset prototypes).

Table 3: Accuracy of $D$ in identifying the correct domain using prototypes from pretrain and target domains.

| Dataset | Known Domain | | Unknown Domain | |
|---|---|---|---|---|
| | Top-1 | Top-5 | Top-1 | Top-5 |
| CompCars | 43.0 | 75.0 | 42.7 −0.3 | 74.6 −0.4 |
| UECFOOD256 | 62.1 | 88.8 | 61.4 −0.7 | 87.7 −1.1 |
| NABirds | 39.1 | 72.8 | 38.5 −0.6 | 71.7 −0.9 |
| Military Aircraft | 12.5 | 33.7 | 8.0 −4.5 | 20.3 −13.4 |
| OxfordPets | 72.3 | 96.6 | 71.8 −0.5 | 96.0 −0.6 |
| Average | 45.8 | 73.4 | 44.5 −1.3 | 70.1 −3.3 |

| Dataset | $\mathcal{D}_p^{d_i}$-Acc | $\mathcal{D}^{d_i}$-Acc |
|---|---|---|
| CompCars | 99.7 | 99.6 |
| UECFOOD256 | 99.5 | 98.5 |
| NABirds | 98.9 | 89.7 |
| Military Aircraft | 60.0 | 95.6 |
| OxfordPets | 99.2 | 97.1 |
| Average | 91.5 | 95.1 |

competitors by more than 8.8% in Top-1 accuracy and 7.8% in Top-5. The FVGAircraft→MilitaryAircraft scenario is the most challenging in our experiments, where results in absolute terms are limited for all approaches, due to the topology of the scenes. The dataset was originally meant for Object Detection, so the target class is often far in the sky, while other confounding objects occupy the foreground. In particular, the highest Top-1 accuracy (12.5%) is achieved by EFFEKT, while the highest Top-5 (39.0%) is achieved by FedAvg+EMA. The last configuration is the simplest, and all methods achieve remarkable accuracy. EFFEKT ranks second very close to FedPromo, achieving a Top-1 accuracy of 72.3% and a Top-5 accuracy of 96.6%. Finally, we remark that EFFEKT closely matches the centralized experiment in most domains.

## 6.2 Multi-Domain Results

Table 2 shows the accuracy achieved in the multi-domain scenario: in the first two columns, clients share domain information to the server, while in the last two columns the information is withheld. The results in Section 6.1 correspond to the case where the sample's domain is known (*i.e.*, when the server-side FM receives a sample, the domain index is included). If this information is not provided, the domain classifier $D$ must be used to estimate it, which may lead to a decrease in accuracy due to domain misclassification. The Table shows that the impact of this issue is limited, with a drop in Top-1 accuracy of less than 1% in all domains except for MilitaryAircraft. For consistency, we also report the Top-5 accuracy, which exhibits a trend similar to that of Top-1. These results are confirmed by Table 3, which reports the domain classification performance. Using the pretraining datasets prototypes, the domain is recognized correctly with an accuracy of 99% for all domains except MilitaryAircraft, where it drops to 60%. This domain proved harder to recognize, as the images are significantly different from those in FGVCAircraft. In many images, the target aircraft appears small and distant, while other objects occupy the foreground. In FGVCAircraft, instead, there is always a plane as the main object. This semantic misalignment is also confirmed by the domain shift analysis provided in Sec. A.5 of the Appendix. In the Table, we also report the accuracy when computing prototypes on the target datasets; in this case the accuracy drop on MilitaryAircraft is restored.

## 6.3 On-device Training Results

Table 4 shows the accuracy achieved by our on-device setup. We performed two different tests, setting the number of active clients $n_a$ to 2 and 4. The results confirm the validity of our simulations, given the tight match between experiments. Further validation experiments are in the Appendix: it shows that the real and simulated validation accuracies match is preserved at all steps of the training evolution (see Fig. A.3). We also measured the power consumption and network usage throughout the federated optimization, as detailed in the Appendix. The former

Table 4: Comparison of on-device and simulated experiments.

| $n_a$ / $n_k$ | Simulation | | Deployment | |
|---|---|---|---|---|
| | Top-1 | Top-5 | Top-1 | Top-5 |
| 2 / 12 | 39.5 | 70.0 | 39.5 | 70.4 |
| 4 / 12 | 40.0 | 70.3 | 39.8 | 70.5 |
| Average | 39.8 | 70.2 | 39.7 | 70.5 |

never exceeds 7W and scales linearly with the number of active clients, while the latter never exceeds 5MB/s. In Figures A.1 and A.2, we show both measures over a short time interval.

# 7 Ablation Studies

**Component Analysis** As a first ablation study, we evaluate the impact of the various components of our architecture on performance using the CompCars dataset. The results are shown in Table 5.

When disabling both C2S and JA Distillations (*i.e.*, using FedAVG for server-side aggregation), the Top-1 accuracy drops by more than 7% (the Top-5 exhibits similar behavior, with a drop of almost 6% as well). The impact of both components on Top-1 accuracy is similar: when using only C2S or only JA, the drop in performance is around 2.5% (from 43% to 40.6% and 40.5%, respectively). This finding demonstrates that both strategies are necessary and complementary; by combining them, we achieve a noticeable improvement. The Top-5 exhibits similar behavior, even though, according to this metric, the C2S strategy has a slightly greater impact. This could stem from the missing cross-entropy objective, which tightens

Table 5: Ablation on the impact of the different modules and losses.

| C2S | JA | | | Top-1 | Top-5 |
| | $\mathcal{L}_{CE}$ | $\mathcal{L}_{KD}^L$ | $\mathcal{L}_{KD}^F$ | | |
|---|---|---|---|---|---|
| ✗ | ✗ | ✗ | ✗ | 35.8 | 69.4 |
| ✗ | ✓ | ✓ | ✓ | 40.6 | 72.6 |
| ✓ | ✗ | ✗ | ✗ | 40.5 | 73.7 |
| ✓ | ✗ | ✓ | ✓ | 41.9 | 74.6 |
| ✓ | ✓ | ✗ | ✓ | 39.2 | 72.7 |
| ✓ | ✓ | ✓ | ✗ | 41.2 | 72.7 |
| ✓ | ✓ | ✓ | ✓ | 43.0 | 75.0 |

the entropy of the predictions, leading to slightly higher Top-1 accuracy at the cost of Top-5. This is confirmed by the ablation on the loss terms used in JA, where disabling the cross-entropy leads to a higher Top-5 accuracy compared to the other partial configurations. Regardless, the experiments confirm the optimal configuration of our approach, as using all components together yields the highest accuracy. To further validate the effectiveness of our two distillation objectives, in Section A.3 of the *Appendix* we report the accuracy when C2S and JA are not performed at every communication round. Overall, we observe an approximately linear increase in performance with respect to the application frequency $f \in [0, 1]$ (one every R rounds). More specifically, the average performance on the server can be estimated as $\text{acc}_1 \simeq 7.93f + 35.3$ and $\text{acc}_5 \simeq 6.05f + 69.1$ with an $R^2$ value of 0.97 for both. As an additional study, we report the comparative analysis when swapping the client encoder architecture to EfficientNet-B0 (Tan & Le, 2020). The results are reported in the Appendix (Figures A.7 for Top-1 Accuracy and A.8 for Top-5 Accuracy), confirming a consistent improvement of EFFEKT with respect to all other competitors (approximately 5% improvement over the second best, FedPromo, and 30% over FedAvg). As a final note, we wish to highlight that distillation on the task-specific pretraining datasets alone is meaningless, as the class-sets are disjoint from those of the clients. Our distillation approaches effectively tackle a data-free Unsupervised Domain Adaptation task on the server (a complementary task to Source-Free UDA Fang et al. (2023), which uses only source data).

**Hyperparameters**
We also evaluate the impact of setting different values for the algorithm hyperparameters. The balance between feature and logits distillation in Eq. 2 is controlled by the $\lambda_{KD}^L$ parameter. As shown in Table 6, the proposed value of 0.01 leads to the best performance, while smaller or larger values result in a decrease in performance for both Top-1 and Top-5 accuracy. Another relevant design parameter is the rank for the LoRA adapters (Table 7). Performance increases when using larger rank adapters up to a certain point, but it decreases again after the optimal value is achieved for rank 16. Additionally, we report the performance achieved by training the full FM without employing adapters (*full FM*). The performance drops below that of rank 32, confirming that the decreasing trend continues to full rank. We hypothesize that this is due to the limited number of samples and the misalignment between proxy and server features, which leads to overfitting using larger adapters.

Table 6: Ablation on logit distillation loss weight.

| $\lambda_{KD}^L$ | Top-1 | Top-5 |
|---|---|---|
| 0.001 | 40.6 | 73.6 |
| 0.005 | 41.5 | 73.8 |
| 0.01 | 43.0 | 75.0 |
| 0.02 | 42.2 | 74.3 |
| 0.1 | 41.7 | 74.5 |

Table 7: Ablation on the rank of LoRA adapters.

| rank | Top-1 | Top-5 |
|---|---|---|
| 8 | 42.0 | 74.2 |
| 16 | 43.0 | 75.0 |
| 32 | 41.5 | 73.9 |
| full FM | 41.3 | 71.1 |

**Federated Configuration**
Here, we evaluate performance in different federated settings. We compared EFFEKT with the baseline FedAvg approach and the best

competitor, *i.e.*, Caligiuri et al. (2025), on the CompCars dataset. We computed results by varying the number of rounds, the Dirichlet concentration parameter, and the total and active client numbers. The detailed results are in the Appendix. Note how our approach consistently outperforms the competitors in all the considered settings. Note that the standard deviation of the Top-1 accuracy of EFFEKT, computed by running five experiments with different random seeds, is very limited (0.5%). This is on par with those of the competitors, in spite of the higher mean values. Figures A.5 and A.6 in the Appendix show how this result is consistent across rounds.

**Out-of-Domain Pretraining Results**

In this section, we examine the performance of EFFEKT when using a large-scale out-of-domain pretraining dataset (ImageNet-1k) in place of the task-specific proxies; the corresponding results are presented in Table 8. Because of the semantic mismatch, the accuracy drops considerably, with EFFEKT being affected more severely than its closest competitor (FedPromo), largely due to the mismatch in data used by server-side distillation. Despite this, our method still ranks second, achieving performance that is very close to the Centralized upper bound. Further details, including per-dataset results, are provided in Section A.3 of the Appendix.

Table 8: Out-of-Domain pretraining results (averaged across domains).

| Method | Top-1 | Top-5 |
|---|---|---|
| FedAvg | 18.8 | 26.3 |
| FedAvg + EMA | 26.2 | 42.6 |
| FedProx | 22.2 | 34.6 |
| MOON | 19.0 | 26.3 |
| FedHEAL | 18.9 | 26.4 |
| FedPromo | **28.5** | **46.4** |
| EFFEKT (ours) | 28.3 | 45.7 |
| Centralized | 28.8 | 46.8 |

## 8 Conclusions

In this paper, we introduced EFFEKT, a novel multi-domain federated learning framework enabling efficient collaboration between lightweight client-side proxy models and a server-side Foundation Model. EFFEKT's key innovation lies in its bi-directional cross-distillation strategies and the efficient server-side training of domain-specific LoRA adapters. Experiments conducted across multiple real-world datasets demonstrated improvements in Top-1 and Top-5 accuracy in most scenarios. Additionally, we validated EFFEKT's real-world applicability through deployments on compute-constrained devices, confirming its efficiency in computation, energy consumption, and network usage. This demonstrated EFFEKT's suitability for privacy-sensitive applications on personal devices, opening up new possibilities for AI in diverse domains.

## Impact Statement

This work aims to advance the field of machine learning and does not introduce immediate or specific societal consequences that require separate emphasis; however, it entails ethical considerations typical of decentralized learning settings. In particular, although raw data remain on client devices, shared model components may still be vulnerable to adversarial attacks that leak sensitive information, underscoring the importance of privacy-preserving techniques. Additionally, decentralized training can encode local biases, which may lead to unfair outcomes if not properly addressed. While the evaluations in this paper are designed to avoid these concerns, caution is still warranted when deploying the proposed methods on sensitive private data, such as facial images, in real-world scenarios. Furthermore, our foundation model optimization approach relies on logits and latents access, which may not always be available, especially when interacting with API-based closed-weight models. Finally, our work requires pretraining data with a reasonable alignment with the target one (as FedPromo), which may not always be available in all domains.

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

# A    Appendix

In the following sections, we report the additional and supporting experiments that could not fit in the main document due to space limitations. We begin by providing experiments varying all aspects of our Federated Learning training configuration in Section A.1; we then move to Section A.2 that reports our power and network consumption measurement setup, as well as our findings; Section A.3 reports some additional quantitative experiments and some training evolution curves, highlighting the stability of out approach; finally, in Section A.8 we report the pseudocode implementation for the remaining modules of EFFEKT.

## A.1    Federated Configuration

This section contains additional experiments varying the Federated Learning configuration, comparing the accuracy of EFFEKT, FedPromo, and FedAvg. In particular, Table A.1 focuses on the number of rounds, Table A.2 on the number of total clients, Table A.3 on the number of active clients, and Table A.4 on the Dirichlet concentration parameter $\alpha$. The results show that the configuration reported in the main document is optimal, leading to the best performance even when changing scenarios, *e.g.*, using fewer clients, or more active clients. We hypothesize that this depends on the bi-directional distillation hyperparameters, which have been tuned with the target configuration.

Regarding the number of rounds, the EFFEKT results behave as expected: the shorter the training, the lower the accuracy. This contrasts with FedPromo, where accuracy remains stable across 250–500 rounds, and with FedAvg, where shorter training actually leads to better performance. The likely reason is that the LoRA adapters in EFFEKT require a relatively larger number of steps to be properly optimized, while the performance drop in FedAvg is caused by client divergence, which FedPromo effectively mitigates. However, further increasing the length of the training beyond 500 rounds does not lead to better performance, *e.g.*, with $n_r = 1000$ rounds the top-1 accuracy is is 42.1% and the top-5 74.9%.

Interestingly, EFFEKT attains higher accuracy even for smaller values of the Dirichlet concentration parameter $\alpha$. This advantage stems from our C2S distillation, which is able to extract useful information from all client updates (including slightly divergent ones). As a result, smaller $\alpha$ values lead to more informative update directions, whereas larger values tend to pull the updates into a similar direction, thereby diminishing their information content. Note how EFFEKT continues to have impressive performances across all settings, as even its lowest accuracy surpasses the highest accuracy achieved by FedPromo. This trend continues at the more extreme values of $\alpha = 0.5$ and 0.1, which yield top-1 41.9/36.7, and top-5 73.3/67.6.

Table A.1: Number of rounds.

| $n_r$ | FedAvg | | FedPromo | | EFFEKT | |
|---|---|---|---|---|---|---|
| | top-1 | top-5 | top-1 | top-5 | top-1 | top-5 |
| 100 | 22.1 | 41.9 | 30.6 | 63.9 | 30.8 | 63.3 |
| 250 | 18.4 | 33.2 | 35.3 | 69.1 | 37.9 | 70.9 |
| 500 | 14.0 | 27.4 | 35.8 | 69.4 | 43.0 | 75.0 |

Table A.2: Number of total clients.

| $n_k$ | FedAvg | | FedPromo | | EFFEKT | |
|---|---|---|---|---|---|---|
| | top-1 | top-5 | top-1 | top-5 | top-1 | top-5 |
| 50 | 10.6 | 22.5 | 36.2 | 68.4 | 40.8 | 72.4 |
| 100 | 14.0 | 27.4 | 35.8 | 69.4 | 43.0 | 75.0 |
| 150 | 12.8 | 24.0 | 29.6 | 59.3 | 39.0 | 70.5 |

Table A.3: Number of active clients.

| $n_a$ | FedAvg | | FedPromo | | EFFEKT | |
|---|---|---|---|---|---|---|
| | top-1 | top-5 | top-1 | top-5 | top-1 | top-5 |
| 5 | 11.0 | 21.9 | 36.9 | 70.1 | 41.6 | 73.3 |
| 10 | 14.0 | 27.4 | 35.8 | 69.4 | 43.0 | 75.0 |
| 20 | 10.8 | 21.5 | 37.2 | 70.7 | 42.1 | 74.4 |

Table A.4: Dirichlet concentration parameter.

| $\alpha$ | FedAvg | | FedPromo | | EFFEKT | |
|---|---|---|---|---|---|---|
| | top-1 | top-5 | top-1 | top-5 | top-1 | top-5 |
| 1 | 14.0 | 27.4 | 35.8 | 69.4 | 43.0 | 75.0 |
| 10 | 12.0 | 24.0 | 37.1 | 70.4 | 42.4 | 74.4 |
| 100 | 12.5 | 24.7 | 37.3 | 70.1 | 42.1 | 74.0 |

## A.2   Power and Network Consumption

As noted in the main document, we deploy EFFEKT on a cluster of 12 compute-constrained real devices, connected to the server and to each other via a LAN network. More specifically, to measure their power consumption, we clustered the clients by manufacturer (Raspberry Pi Foundation vs. NVIDIA), plugging each group into a different Sonoff S31 (Shenzhen Sonoff Technologies Co., Ltd., 2015) smart power plug (running TASMOTA firmware (Tasmota maintainers and contributors, 2017)). In the first group, we included 3 Raspberry Pi 4 and 3 Raspberry Pi 5; in the second group, we had 5 Jetson Nano and 1 Jetson Orin-Nano. Logging was performed using an open source MQTT broker (Mosquitto (Eclipse Foundation, 2009)) and a Python script that polls the server every second. Regarding the network statistics, to avoid affecting the measurements, we deployed a Grafana dashboard (Grafana Labs, 2013) on a separate machine (distinct from the server and clients) that receives information from a Prometheus service (Prometheus maintainers and contributors, 2013) running on each client via a separate network interface.

We measure an average power consumption across the federated optimization of 2.1W and 3.6W by the Raspberry Pi devices, under $n_a = 2$ and $n_a = 4$, respectively. The NVIDIA boards absorb approximately 70% more power, landing at 3.3W and 6.3W under the same configurations. In Figure A.1, we plot the consumption on both configurations over a small time slice. The spikiness of the plot is a consequence of the active client sampling of the server. A full training with 2 active clients absorbs $\sim 3.5$kWh, split in 1.5kWh for Jetson and 2kWh for Raspberry (sampled more often, as they are faster).

A similar plot, showing the network usage over a time window, is reported in Figure A.2. Most of the clients' throughput is in downloads since they need to update the full local model (encoder and head) at each round, but they only send the head to the server. This is confirmed by the global statistics, which never exceed 4MB/s in RX, and 60kB/s in TX. Moreover, on average each client transmits 3.7MB (which closely matches the size of $H_k^{d_i}$) and receives 47.9MB.

## A.3   Additional Experiments

Figures A.3 and A.4 report the Top-1 and Top-5 accuracy achieved by EFFEKT on the CompCars dataset under configurations with 2 and 4 active clients, comparing simulated experiments against real-device deployments. The results obtained on real hardware closely match those observed in simulation, providing strong evidence for the reliability of our simulated setup. As expected, increasing the number of active clients from 2 to 4 leads to higher accuracy; however, the performance gap remains relatively small. This limited difference highlights the robustness and stability of EFFEKT, which is enabled by its bidirectional distillation mechanism and allows the framework to perform consistently even with fewer participating clients.

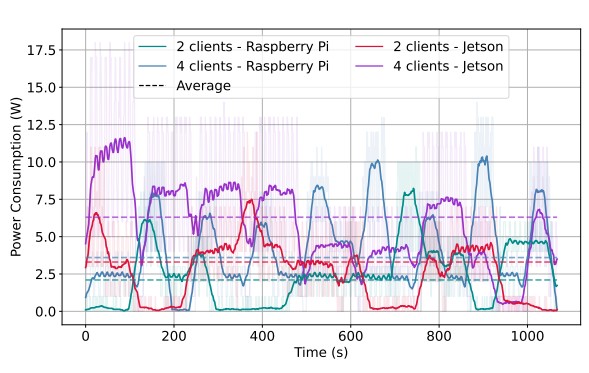

Figure A.1: Training power consumption (short time window).

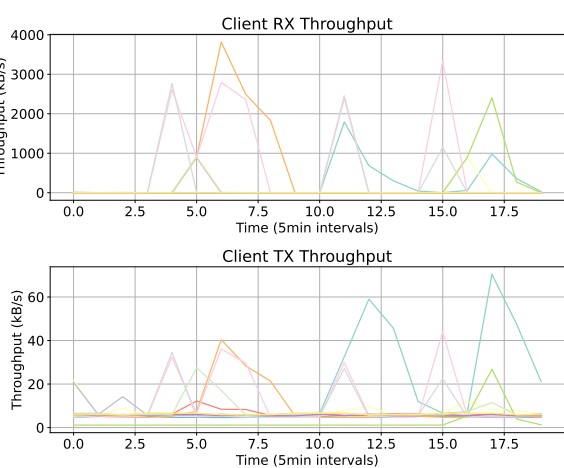

Figure A.2: Training network usage (short time window).

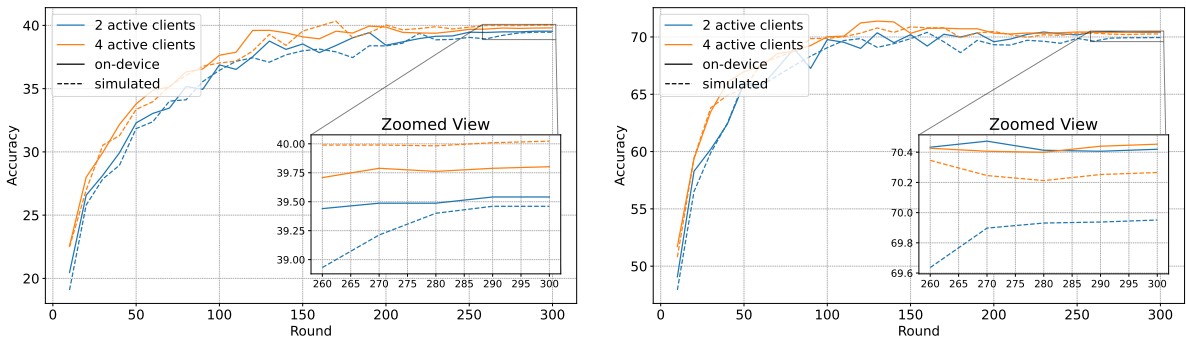

Figure A.3: Top-1 acc. on simulated and real devices. Figure A.4: Top-5 acc. on simulated and real devices.

Furthermore, Figures A.5 and A.6 illustrate the training evolution of Top-1 and Top-5 accuracy across communication rounds, including a shaded region representing the confidence interval computed over 5 independent runs with different random seeds. EFFEKT consistently outperforms all competing methods throughout training, exceeding the closest baseline, FedPromo, by more than 5% in both metrics. Notably, the confidence intervals (for both Top-1 and Top-5) of EFFEKT never overlap with those of the other approaches, providing clear evidence that the observed improvements are statistically significant and robust.

As additional validation, in Figures A.7 and A.8 we analyze training dynamics when changing the client-side architecture, from MobileNetV3-Small to EfficientNet-B0, and compare EFFEKT with FedPromo and FedAvg.

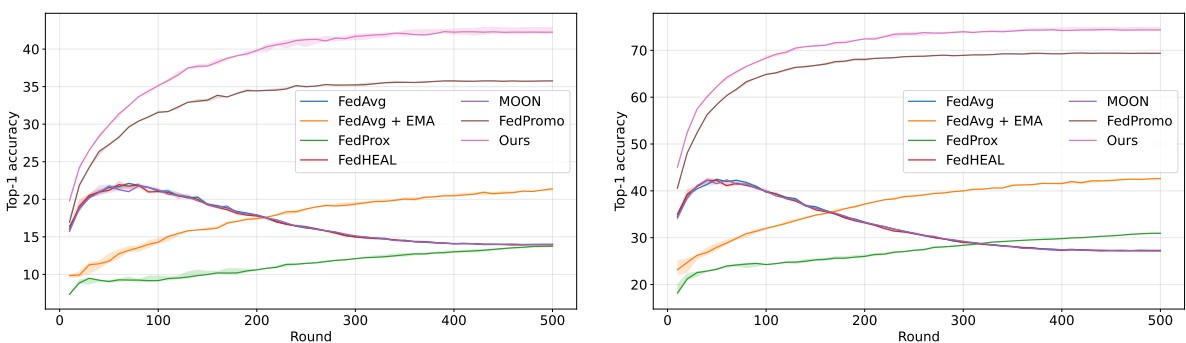

Figure A.5: Validation server top-1 accuracy evolution.

Figure A.6: Validation server top-5 accuracy evolution.

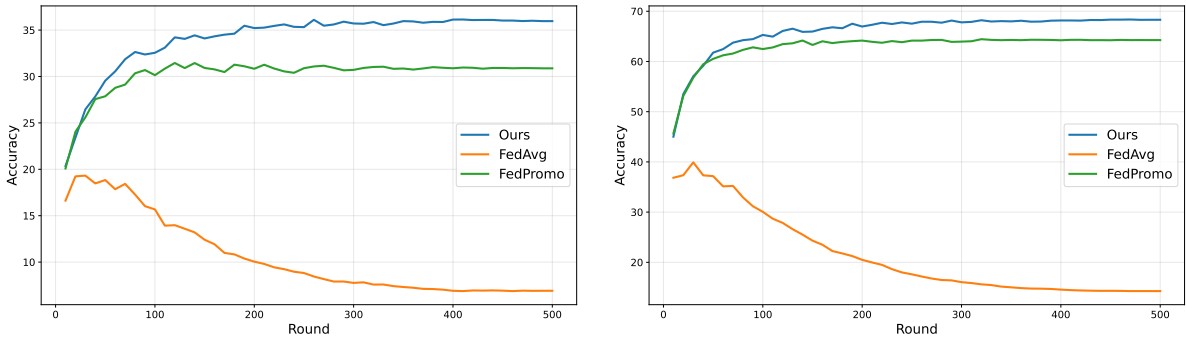

Figure A.7: Validation server top-1 accuracy evolution using EfficientNet-B0 as client encoder.

Figure A.8: Validation server top-5 accuracy evolution using EfficientNet-B0 as client encoder.

In this setting, FedAvg collapses in both Top-1 and Top-5 accuracy, whereas FedPromo and EFFEKT remain stable across training rounds. Despite EfficientNet-B0 having a larger number of parameters and relying on a different architectural design compared to the standard classification head, overall performance is lower than in previous experiments.

Nevertheless, EFFEKT maintains a clear advantage, achieving approximately a 5% improvement over FedPromo in both accuracy metrics, further demonstrating its robustness to architectural changes on the client side.

In Table A.5, we report results analogous to those presented in Table 2 of the main paper, with the key difference that the prototypes of the domain classifier are computed using the private target-domain data rather than the task-specific pretraining datasets.

Overall, the results closely match those obtained in the main table, thereby further validating the effectiveness and consistency of the proposed approach. The only notable deviation is observed on the MilitaryAircraft dataset, where performance improves substantially. This gain can be attributed to the removal of the domain shift between MilitaryAircraft and its previously coupled domain, FGV-CAircraft, when prototypes are derived directly from the target data. It is important to emphasize, however, that these results should be interpreted strictly as upper bounds, since in a realistic federated deployment, the server does not have access to the clients' private target-domain data.

Furthermore, Table A.6 presents companion results to Table 1 in the main document, obtained by replacing the multiple task-specific pretraining datasets with a single generic dataset, ImageNet. In this setting, performances degrade significantly across most domains. This drop highlights that the client-side encoder alone is not sufficiently expressive to accurately approximate the server-side foundation model, leading to increased representation misalignment during federated training. Despite this limitation, EFFEKT achieves performances that are largely comparable to both FedPromo and centralized training in most configurations. Absolute accuracy remains

Table A.5: Accuracy on multi-domain scenario. *Domain Knowledge*: query sample includes domain index $i$. *Unknown Domain: $D$* used to estimate $i$.

| Dataset | Domain Knowledge | | Unknown Domain | |
|---|---|---|---|---|
| | Top-1 | Top-5 | Top-1 | Top-5 |
| CompCars | 43.0 | 75.0 | 42.7 $-0.3$ | 74.6 $-0.4$ |
| UECFOOD256 | 62.1 | 88.8 | 60.6 $-1.5$ | 86.9 $-1.9$ |
| NABirds | 39.1 | 72.8 | 33.7 $-5.4$ | 62.7 $-10.1$ |
| Military Aircraft | 12.5 | 33.7 | 11.8 $-0.7$ | 31.7 $-2.0$ |
| OxfordPets | 72.3 | 96.6 | 70.5 $-1.8$ | 94.0 $-2.6$ |
| Average | 45.8 | 73.4 | 43.9 $-1.9$ | 70.0 $-3.4$ |

Table A.6: Out-of-domain experiments. Federated Top-1 and Top-5 accuracy ($\uparrow$) starting from ImageNet-1k (1000 classes) pretraining. **Best** in bold, second-best underlined.

| Dataset $\mathcal{D}$ | Method | DINOv2 | |
|---|---|---|---|
| | | Top-1 | Top-5 |
| CompCars | FedAvg | 1.2 | 3.9 |
| | FedAvg + EMA | 3.0 | 9.7 |
| | FedProx | 2.4 | 8.5 |
| | MOON | 1.3 | 3.9 |
| | FedHEAL | 1.2 | 4.0 |
| | FedPromo | **3.3** | **10.8** |
| | EFFEKT (ours) | 2.9 | 10.1 |
| | Centralized | 3.3 | 9.8 |
| UECFOOD256 | FedAvg | 2.1 | 6.9 |
| | FedAvg + EMA | 23.5 | 48.4 |
| | FedProx | 4.0 | 13.2 |
| | MOON | 2.0 | 7.1 |
| | FedHEAL | 2.1 | 7.0 |
| | FedPromo | 24.0 | 48.5 |
| | EFFEKT (ours) | **24.9** | **49.4** |
| | Centralized | 26.1 | 52.5 |
| NABirds | FedAvg | 1.5 | 5.1 |
| | FedAvg + EMA | 11.3 | 29.1 |
| | FedProx | 10.5 | 26.6 |
| | MOON | 1.6 | 5.0 |
| | FedHEAL | 1.4 | 4.9 |
| | FedPromo | **18.5** | **48.6** |
| | EFFEKT (ours) | 16.5 | 44.5 |
| | Centralized | 16.7 | 43.0 |
| Military Aircraft | FedAvg | 5.1 | 16.2 |
| | FedAvg + EMA | 8.0 | **25.8** |
| | FedProx | **8.3** | 24.8 |
| | MOON | 5.3 | 16.2 |
| | FedHEAL | 5.1 | 16.5 |
| | FedPromo | 7.1 | 24.1 |
| | EFFEKT (ours) | 8.0 | 24.6 |
| | Centralized | 9.2 | 28.8 |
| OxfordPets | FedAvg | 84.0 | 99.5 |
| | FedAvg + EMA | 85.2 | 99.8 |
| | FedProx | 85.6 | **99.9** |
| | MOON | 84.7 | 99.5 |
| | FedHEAL | 84.5 | 99.5 |
| | FedPromo | **89.5** | 99.8 |
| | EFFEKT (ours) | 89.0 | **99.9** |
| | Centralized | 88.5 | 99.9 |

low for nearly all domains, with the notable exception of OxfordPets, which is effectively task-specific with respect to ImageNet, as its samples are drawn from the same validation set.

Finally, in Table A.7, we report the final server accuracy when our two distillation techniques (C2S and JA) are not performed at every communication round. The results suggest a roughly linear relationship between the frequency of the distillation and the server performance. More specifically, we report the performance when C2S and JA are enabled with frequency $f = \{1, \frac{1}{2}, \frac{1}{5}, \frac{1}{10}, 0\}$ rounds. Linear fitting attains the following relationship between the two metrics: $acc_1 \simeq 7.93f + 35.3$ and $acc_5 \simeq 6.05f + 69.1$ with an $R^2$ value of 0.97 for both.

| Frequency | Top-1 | Top-5 |
|---|---|---|
| 0 (FedPromo) | 35.8 | 69.4 |
| 1/10 | 35.4 | 69.0 |
| 1/5 | 36.6 | 70.5 |
| 1/2 | 40.0 | 72.5 |
| 1 (EFFEKT) | 43.0 | 75.0 |

Table A.7: Accuracy vs. distillation frequency.

## A.4 Privacy Considerations

In this section, we provide some preliminary results applying Local Differential Privacy with norm clipping to our proposed aggregation scheme. For fair comparison, we follow the same configuration as Caligiuri et al. (2025). The results are reported in Table A.8, where we compare the performance of FedPromo and EFFEKT across a wide range of $\varepsilon$ values after fixing $\delta = 10^{-5}$ on the CompCars dataset. Overall, the performance consistently surpasses that of FedPromo and only begins to noticeably deteriorate when very small values of $\varepsilon$ are used ($\varepsilon = 2.5, 1$). Note that, in this setting, the privacy guarantor is based on the moments accountant (Abadi et al., 2016) with composition over $n_r = 500$ rounds and the Poisson subsampling amplification from the active-client sampling ($n_a/n_k = 0.1$)

| $\varepsilon$ | FedPromo | | EFFEKT | |
|---|---|---|---|---|
| | top-1 | top-5 | top-1 | top-5 |
| ✗ | 35.8 | 69.4 | 43.0 | 75.0 |
| 50 | 34.7 | 67.8 | 39.6 | 71.5 |
| 10 | 33.6 | 66.6 | 39.0 | 70.7 |
| 5 | 31.0 | 62.4 | 35.6 | 66.5 |
| 2.5 | 24.7 | 54.0 | 27.5 | 56.2 |
| 1 | 14.2 | 33.7 | 15.1 | 33.9 |

Table A.8: Accuracy with LocalDP.

Another theoretical privacy guarantee is provided by the limited number of trainable parameters on the clients, more specifically, theorem 4.1 in Zhang & Chen (2024). In our case, the limiting factor of the rank of $\mathbf{J} \in \mathbb{R}^{d \times (B \cdot p)}$ is the number of parameters $d$, as batch size $B$ and input dimensionality $p$ are large.

## A.5 Domain Shift Study

We employed the CLIP-I and DINO metrics (Ruiz et al., 2023) computed between dataset prototypes (*i.e.*, the mean feature vectors over all samples in each dataset) to quantify domain shift across datasets. The corresponding results are shown in Figure A.9. The DINO metric focuses on the semantic content of images, providing a clear measure of domain shift; conversely, the CLIP-I metric is more sensitive to stylistic properties, which may lead to inflated scores, as all images are real-world photographs without additional stylization.

From the figure, we observe that the pre-training datasets exhibit reasonably good, though heterogeneous, alignment with their respective target datasets. At the same time, their content differs, with an average (DINO) cosine similarity of about 0.67. Notably, the two aircraft datasets show weaker alignment than the others, which partially accounts for the lower performance in this scenario. The CLIP-I metric broadly tracks the behavior of DINO but yields consistently higher minimum similarity values. This effect arises from the fact that all images are real-world, which biases CLIP-I toward higher similarity scores.

## A.6 Server-Side Compute Costs

In this section, we report additional information on the computational cost (wall-clock time and maximum VRAM) incurred by the Server when running the EFFEKT pipeline.

Regarding wall-clock time, the server-side overhead introduced by C2S and JA is moderate and scales predictably with distillation frequency (one-every-x rounds, 1:x). Note that all measurements reported in the following include also the cost of simulating $n_a = 10$ active clients for each of the $n_r = 500$ rounds.

Starting from a baseline of 9h (no C2S/JA, *i.e.*, only clients), adding server-side distillation increases training time to 11.1h, 11.7h, and 13.2h for 1:10, 1:5, and 1:2 distillation ratios, respectively. The full EFFEKT

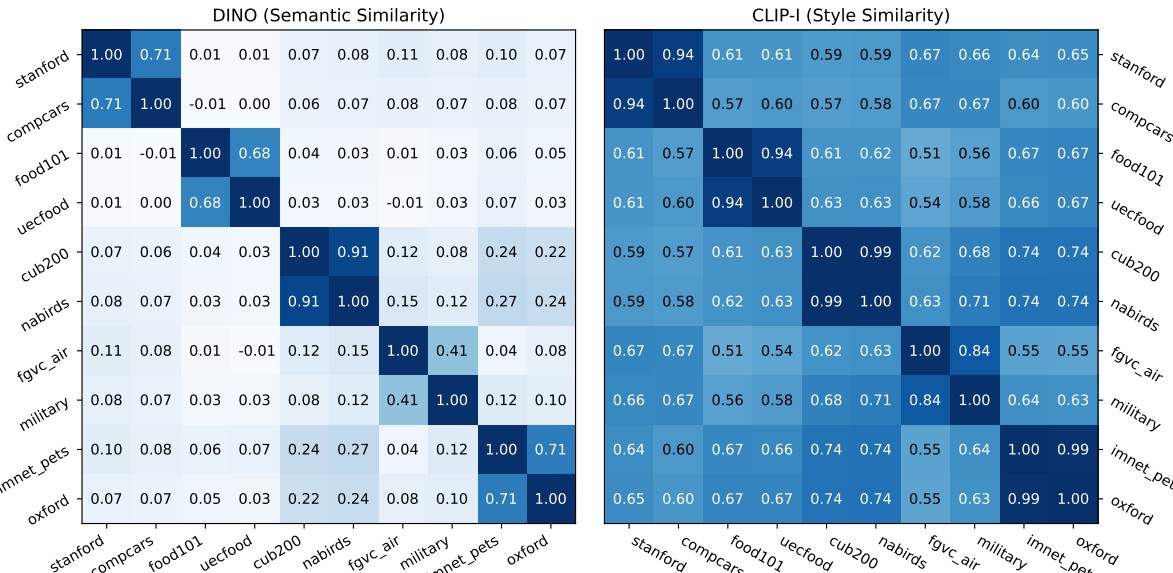

Figure A.9: DINO and CLIP-I metrics.

configuration requires 17h, which rises to 21.3h when replacing LoRA with full DINO fine-tuning, while ablated variants (EFFEKT no JA, EFFEKT no C2S) fall at 15.4h and 13.2h, respectively. From these results, we can estimate an average additional cost of 1.5m/round when C2S and JA are enabled (with C2S accounting for 60.4% of the additional cost and JA the remaining 39.6%).

Regarding GPU memory (VRAM), the footprint remains approximately constant across C2S/JA configurations, since the classifiers involved in C2S are negligible in size compared to the DINO encoder. The more meaningful differences emerge when comparing optimization strategies: FedPromo requires $\sim$ 11GB, EFFEKT $\sim$ 14GB (thanks to the parameter efficiency of LoRA adapters), and full fine-tuning of the DINO encoder $\sim$ 19GB (all at batch size 64). Overall, these results support the efficiency claim of EFFEKT, demonstrating a favorable trade-off between server-side cost and performance gains.

## A.7 Statistical Analysis

As supporting evidence for the results reported in Figures A.5 and A.6, we also computed the average and standard deviation of the final-round top-1 and top-5 accuracy. The results are reported in Table A.9, and they confirm the stability of EFFEKT. More specifically, across all datasets and metrics, our method enjoys very tight bounds on the distribution. The standard deviation never exceeds the 1% mark, except for the top-1 accuracy in the Oxford-Pets dataset, confirming that the gains reported in the experimental evaluation are statistically significant.

Table A.9: Mean $\mu$ and Standard Deviation $\sigma$ of the top-1 and top-5 accuracy of the final round.

| Dataset | top-1 | | top-5 | |
|---|---|---|---|---|
| | $\mu$ | $\sigma$ | $\mu$ | $\sigma$ |
| CompCars | 42.4 | 0.5 | 74.6 | 0.4 |
| UECFOOD256 | 62.0 | 0.1 | 88.9 | 0.1 |
| NABirds | 39.4 | 0.3 | 72.8 | 0.2 |
| MilitaryAircraft | 12.2 | 0.5 | 33.7 | 0.2 |
| OxfordPets | 73.8 | 1.5 | 96.9 | 0.3 |

## A.8 EFFEKT Pseudocode

In this section, we report the pseudocode implementation of EFFEKT. We begin by detailing the EFFEKT domain identification pipeline in Algorithm A.1. We then report a detailed description of the EFFEKT ClientRound, implementing the ICP regularization (Caligiuri et al., 2025), in Algorithm A.2, and of our two main components (C2S and JA) in Algorithms A.3 and A.4.

---

**Algorithm A.1** EFFEKT Multi-Domain Server Inference.

---

**Require:** Server encoder $\hat{O}$, domain classifier $D$, domain LoRAs $\{L^{d_i}\}_{i \in \mathcal{I}}$, domain heads $\{H_k^{d_i}\}_{i \in \mathcal{I}}$
**Input:** Input image $\mathbf{X}$
 1: $i \leftarrow D(\mathbf{X})$                                     // Compute domain index
 2: $O^{d_i} \leftarrow \text{ApplyLoRA}\left(\hat{O}, L^{d_i}\right)$                           // Apply correct LoRA
 3: $\mathbf{F}^O \leftarrow O^{d_i}(\mathbf{X})$                                 // Compute features
 4: $\mathbf{L}^O \leftarrow H^{d_i}(\mathbf{X})$                                // Compute logits
 5: $\tilde{y} \leftarrow \text{ArgMax}_{y \in \mathcal{Y}}(\mathbf{L}^O[y])$                   // Compute prediction
**Output:** $\tilde{y}$

---

**Algorithm A.2** EFFEKT ClientRound.

---

**Require:** Number of local epochs $n_e$, domain index $i$, client index $j$, batch size $b$, encoder $E^{d_i}$
**Input:** Previous round's head $H_k^{d_i}$, and current $lr$
 1: Initialize local model $M_{k_j}^{d_i} \leftarrow H_k^{d_i} \circ E^{d_i}$
 2: **for** $e \leftarrow 1 \ldots n_e$ **do**
 3:     $M_{k_j}^{d_i} \leftarrow \text{epochICP}\left(M_{k_j}^{d_i}, b, lr, e\right)$              // See Caligiuri et al. (2025) for details
 4: **end for**
 5: Extract current head $H_{k_j}^{d_i}$ from $M_{k_j}^{d_i}$
**Send:** $H_{k_j}^{d_i}$ to server

---

**Algorithm A.3** EFFEKT Clients-to-Server distillation.

---

**Require:** Domain index $i$, server encoder $\hat{O}$, pretraining dataset $\mathcal{D}_p^{d_i}$, batch size $b$
**Input:** LoRA $L^{d_i}$, previous round's encoder $E^{d_i}$, updated heads from clients $\mathcal{K}_a^{d_i}$: $\mathcal{H}^{d_i} = \{H_{k_j}^{d_i}\}_{j \in \mathcal{K}_a^{d_i}}$
 1: $O^{d_i} \leftarrow \text{ApplyLoRA}\left(\hat{O}, L^{d_i}\right)$
 2: **for** $q \leftarrow 1 \ldots \lfloor |\mathcal{D}_p^{d_i}|/b \rfloor$ **do**
 3:     Extract $b$ samples from dataset $\mathcal{D}_p^{d_i}$ into $\mathcal{B}$
 4:     $l \leftarrow 0$                                      // Initialize batch loss
 5:     **for** $(\mathbf{X}, y) \in \mathcal{B}$ **do**
 6:         $\mathbf{F}^O \leftarrow O^{d_i}(\mathbf{X})$                          // Extract Features
 7:         $\mathbf{F}^E \leftarrow E^{d_i}(\mathbf{X})$
 8:         **for** $H^{d_i} \in \mathcal{H}^{d_i}$ **do**
 9:             $\mathbf{L}^O \leftarrow H^{d_i}(\mathbf{F}_O)$                    // Compute Logits
10:             $\mathbf{L}^E \leftarrow H^{d_i}(\mathbf{F}_E)$
11:             $l \leftarrow l + \mathcal{L}_{KD}^L(\mathbf{L}_O, \mathbf{L}_E)$           // Accumulate loss
12:         **end for**
13:     **end for**
14:     $L^{d_i} \leftarrow \text{AdamOptim}(L^{d_i}, l)$                   // Update LoRA
15: **end for**
**Output:** $L^{d_i}$

---

---

**Algorithm A.4** EFFEKT Joint Alignment distillation.

---

**Require:** Domain index $i$, server encoder $\hat{O}$, pretraining dataset $\mathcal{D}_p^{d_i}$, batch size $b$, logit KD loss weight $\lambda_{KD}^L$
**Input:** Aggregated client model $M_k^{d_i}$, LoRA $L^{d_i}$, pretraining head $H_p^{d_i}$

1:   $O^{d_i} \leftarrow \text{ApplyLoRA}\left(\hat{O}, L^{d_i}\right)$
2:   $H_k^{d_i} \circ E^{d_i} = M_k^{d_i}$                 // Split model in encoder and head
3:   **for** $q \leftarrow 1 \dots \lfloor |\mathcal{D}_p^{d_i}|/b \rfloor$ **do**
4:      Extract $b$ samples from dataset $\mathcal{D}_p^{d_i}$ into $\mathcal{B}$
5:      $l_{KD}^L \leftarrow 0$                 // Initialize batch logits KD loss
6:      $l_{KD}^F \leftarrow 0$                 // Initialize batch feature KD loss
7:      $l_{CE} \leftarrow 0$                 // Initialize batch cross entropy loss
8:      **for** $(\mathbf{X}, y) \in \mathcal{B}$ **do**
9:         $\mathbf{F}^O \leftarrow O^{d_i}(\mathbf{X})$            // Extract features
10:        $\mathbf{F}^E \leftarrow E^{d_i}(\mathbf{X})$
11:        $\mathbf{L}^O \leftarrow H_k^{d_i}(\mathbf{F}^O)$           // Client head logits
12:        $\mathbf{L}^E \leftarrow H_k^{d_i}(\mathbf{F}^E)$
13:        $\mathbf{L}_p^O \leftarrow H_p^{d_i}(\mathbf{F}^O)$          // Pretraining head logits
14:        $\mathbf{L}_p^E \leftarrow H_p^{d_i}(\mathbf{F}^E)$
15:        $l_{KD}^F \leftarrow l_{KD}^F + \mathcal{L}_{KD}^F(\mathbf{F}^O, \mathbf{F}^E) + \mathcal{L}_{KD}^F(\mathbf{F}^E, \mathbf{F}^O)$
16:        $l_{KD}^L \leftarrow l_{KD}^L + \mathcal{L}_{KD}^L(\mathbf{L}^O, \mathbf{L}^E) + \mathcal{L}_{KD}^L(\mathbf{L}^E, \mathbf{L}^O)$
17:        $l_{CE} \leftarrow l_{CE} + \mathcal{L}_{CE}(\mathbf{L}_p^O, y) + \mathcal{L}_{CE}(\mathbf{L}_p^E, y)$
18:      **end for**
19:      $l \leftarrow l_{CE} + l_{KD}^F + \lambda_{KD}^L \cdot l_{KD}^L$       // Merge losses
20:      $M_k^{d_i} \leftarrow H_k^{d_i} \circ E^{d_i}$          // Merge client model
21:      $\left(L^{d_i}, M_k^{d_i}, H_p^{d_i}\right) \leftarrow \text{AdamOptim}(L^{d_i}, M_k^{d_i}, H_p^{d_i}, l)$    // Update Models
22: **end for**
**Output:** $\left(M_k^{d_i}, L^{d_i}, H_p^{d_i}\right)$

---

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
