# OpenReview forum: "EFFEKT: Efficient Federated Knowledge Transfer to Foundation Models"
_TMLR — Under review for TMLR_

### Review · Reviewer_sLT4 · 2026-06-03

**Summary Of Contributions:**

The paper proposes an efficient federated knowledge transfer solution to distill knowledge from LoRA modules into the server model. Experiments on real devices and in real-world environments demonstrate the performance of the proposed method.

Strengths:

1. The authors evaluate the algorithm in real federated learning environments on resource-constrained devices.
2. The federated learning setting is reasonable, with 100 clients and 10 clients participating in each round.
3. The paper provides clear ablations on hyperparameter settings and component analysis.

Weaknesses:

1. “Consider a scenario where a technology company aims to offer a personalized remote image recognition service exploiting a powerful foundation model at the server side.” This scenario does not make sense to me. We do not need a foundation model for image recognition tasks. The provided examples (“e.g., plant or insect species recognition, or vehicle model identification”) can be easily achieved using a lightweight CNN and personalized federated learning. I do not understand the issue with simple averaging of network weights at the server side.
2. I also do not understand this motivation: “In the context of our earlier example, it allows for more effective learning of new, previously unknown classes at the server level without direct access to client data.” Why can previous methods, such as meta-federated learning, not achieve this?
3. Figure 2 is confusing, as the workflow contains multiple rounds of server-client communication. How is latency reduced? Why does the server need access to label information in (4) Joint Alignment? Would this not break the privacy protocol?
4. In Section 3, H and E are not defined. What is “domain-asynchronous training”?
5. MobileNetV3-small is not strictly a foundation model.
6. The motivation for using the “selective weight optimization strategy” does not make sense.
7. In client-to-server distillation, if the server needs access to the logits, would that not leak privacy?
8. An (\alpha = 1) Dirichlet distribution only represents moderate concentration. The authors should also evaluate smaller values.
9. How do the authors guarantee convergence of the distillation method? The curves in Figures A.5 and A.6 do not appear to be fully converged.

Minor Comments:

1. In Figure 1, what does “C2S Distill.” mean? Figure 2 also contains many notations and terminologies that are not introduced.

**Audience:**

No

**Audience Explanation:**

The major issue is that the paper's motivation is not clear. It proposes a new method on a well-explored problems and the reason to adopt an alternative does not make sense. The authors need to give clear definition of the terminologies and adequately emphasize the reason that why we need such a method.

**Broader Impact Concerns:**

No ethical concerns. The authors are encouraged to add one section on discussing broader impact.

**Claims And Evidence:**

No

**Claims Explanation:**

The entire methodology and framing of the research problem sound like a random combination of several irrelevant concepts. Each choice does not support the others, and the overall flow lacks rationale. There are many undefined terminologies and misused concepts that the authors need to address more carefully. Considering the challenges of deploying foundation models in the real world while allowing each client to learn domain-specific knowledge, I think there is merit in deploying multiple clients in real hardware experiments; however, the authors do not fully demonstrate the rationale for doing so.

**Requested Changes:**

1. Give clear rationale and motivation why such methods are needed (see my comments in weakness).
2. Many terminologies are not defined and many concepts are misused or confusing, please revise it.
3. Analysis that privacy protocol will not be violated.
4. Experiments with smaller alpha.
5. Convergence analysis.

---

> ### Author Response · Authors · 2026-06-04
> **Response to sLT4 - Part 1/2**
>
> **SUMMARY**\
> Knowledge transfer in C2S is **from** client heads **to** server-side LoRA adapters (Section 4, Eq. 1). LoRAs are **only on the server**, trained to adapt a frozen server-side FM to client domains. Clients optimize a classifier on frozen CNN-based encoders pre-aligned with the FM. The server never sees client data; knowledge transfer occurs only through federated optimization of classification heads.
>
> Motivation breakdown:
> 1. **Problem:** FMs are powerful but cannot be deployed on edge devices, yet standard FL requires client-side training.
> 2. **Solution 1:** Lightweight proxy models with aligned feature spaces for clients (Section 4).
> 3. **Solution 2:** C2S distillation transfers client knowledge to server LoRA adapters *without accessing private data* (Section 4.1).
> 4. **Solution 3:** JA re-aligns proxy and FM feature spaces using public pretraining data (Section 4.1).
>
> **W1**\
> We consider fine-grained classification, where small CNNs are insufficient. MobileNetV3-small and DINOv2 (ViT-L/14-Reg) differ drastically: ImageNet-1k accuracy is $67.7\%$ vs. $86.5\%$ (~$20\%$ gap) [R2, R1]. The objective is to teach client domains to the server FM **without the server seeing any private client data**, while the FM is too large for client deployment — ruling out standard FL. We align CNN proxy feature spaces with the FM and propagate federated knowledge via server-side LoRAs. FedAvg performs on average $24.4\%$ lower in top-1 and $33.9\%$ lower in top-5 than EFFEKT (Table 1).
>
> **W2**\
> No Meta-FL methods target CA-FKT. Meta-FL, like standard FL, requires deploying the target model on clients, sharing the same limitation. The CA-FKT task involves: (1) heterogeneous client/server architectures, (2) domain-asynchronous training across non-overlapping client groups, and (3) privacy-preserving knowledge transfer to a server-side FM. Meta-FL (Aramoon et al., 2021) only shares client clustering; it does not address client-server architectural mismatch, nor do our approach uses SecAgg (Bonawitz et al., 2017), which is a core feature of Meta-FL.
>
> **W3**\
> Figure 2 shows 4 stages: a) feature-alignment pretraining (server-side); b) local client updates (head training, encoder frozen); c) C2S distillation (server-side, using pretraining data); d) JA (server-side, re-aligns client encoder to FM space before next round).\
> Clients upload only the small classification head ($3.7$MB vs. $47.9$MB for the full encoder). **The server uses only labels from its public pretraining data, not private client data.** This is stated in Section 4 (JA paragraph) and confirmed by Algorithm A.4. No private client data, labels, or raw samples are ever transmitted to the server.
>
> **W4**\
> $H$ and $E$ are defined in Section 3, last line of the third paragraph. We acknowledge "domain-asynchronous training" lacks a standalone definition and will add one: it refers to the scenario where clients from different domains participate in different FL rounds without requiring all domains to be active simultaneously, allowing the server to update domain-specific LoRA adapters independently.
>
> **W5**\
> **MobileNetV3-small is not a foundation model** — that is precisely why we use it. It is the *client-side proxy model* (Section 4: *"the client model is a MobileNetV3-small with a linear feature translator module"*). The server-side FM is DINOv2 (ViT-L/14-Reg). The distinction between the lightweight client proxy and the server-side FM is the central architectural premise of the paper.
>
> **W6**\
> We use ICP as the strongest available distributed optimization strategy. ICP is introduced alongside the CA-FKT task in [R3], enabling a fair comparison between simple averaging (ICP alone) and EFFEKT (ICP + C2S + JA). Confirming compatibility with distributed optimization is a strength of the approach.
>
> **W7**\
> The server computes all logits on its own public pretraining dataset $D^{d_i}_p$ using the uploaded frozen client head weights. No client logits are transmitted. Specifically:
> 1. Clients upload only trained head weights $H^{d_i}_{k_j}$ (no raw data, no private labels).
> 2. The server feeds public pretraining samples through both encoders, obtaining features $F^O$ and $F^E$.
> 3. Logits are computed server-side by applying the uploaded frozen head to server-computed features.\
> This is consistent with standard FL privacy assumptions and explicitly stated in Section 4.
>
> **W8**\
> $\alpha=1$ is already considered heterogeneous (Hsu et al., 2019) and is the standard value in prior FL literature including [R3]. Additional experiments with $\alpha=0.5$ and $\alpha=0.1$ yield: top-1 $41.9\%$ / $36.7\%$, top-5 $73.3\%$ / $67.6\%$, respectively — similar to the $\alpha=1$ case.

---

> > ### Author Response · Authors · 2026-06-04
> > **Response to sLT4 - Part 2/2**
> >
> > **W9**\
> > Figures A.5–A.6 show EFFEKT's accuracy approaching a stable plateau by round 500 with narrowing confidence intervals. Figures A.3–A.4 confirm stability in final rounds. A formal convergence proof is not possible in this setting, consistent with most FL papers (FedAvg, MOON, FedHEAL, FedPromo). Standard deviation across 5 runs is $0.5\%$ (Section 7); Table A.1 shows monotonically improving performance. An additional experiment with $n_r=1000$ rounds attains $42.1\%$ top-1 and $74.9\%$ top-5, both within $1\%$ of the 500-round score.
> >
> > **MC1**\
> > C2S Distill. = *Clients-to-Server distillation*, defined in Section 4.1. We will update the figure caption to define the acronym and move Figure 2 after Section 3. All symbols are defined in Section 3; we could not identify any missing definition — could the reviewer provide details?
> >
> > **SUBMISSION CLAIMS**\
> > We respectfully but firmly disagree. The pipeline is coherent and tightly motivated, with each component directly enabling the next:
> > 1. **Pretraining** (Section 4): Aligns the lightweight proxy feature space to the FM, enabling cross-architecture knowledge transfer.
> > 2. **Client local training with ICP** (Section 4.1, Alg. A.2): Trains only head $H^{d_i}_{k_j}$ on private data, preserving encoder alignment across clients and rounds.
> > 3. **C2S distillation** (Section 4.1, Alg. A.3): Transfers client-learned knowledge to server LoRA adapters *without accessing private data*, updating only 1.57M parameters ($\sim194\times$ less than the full FM, Section 5).
> > 4. **JA distillation** (Section 4.1, Alg. A.4): Re-aligns proxy and FM feature spaces after the LoRA update, ensuring the next round's clients start from a well-aligned initialization.\
> > Ablation (Table 5): removing C2S or JA alone causes ~$2.5\%$ top-1 drop ($43.0\% \rightarrow 40.5\%$ / $40.6\%$); removing both reduces to FedPromo baseline ($35.8\%$). The pipeline is validated across 5 fine-grained datasets (Table 1), multiple federated configurations (Tables A.1–A.4), and 12 real edge devices (Table 4).
> >
> > **AUDIENCE INTEREST**\
> > Federated learning with FMs is highly active, as evidenced by *"Advances and Open Challenges in Federated Foundation Models"* (Ren et al., 2025) cited in our Related Work. Real-device validation on heterogeneous low-power hardware is rare in FL. We provide energy consumption (never exceeding 7W, Appendix A.2), network bandwidth (never exceeding 5MB/s, Section 6.3) across 300 rounds on 12 physical devices (3× RPi4, 3× RPi5, 5× Jetson Nano, 1× Jetson Orin-Nano), with simulated and real-device results closely matching (Table 4, Figures A.3–A.4).
> >
> > **C1**\
> > See replies to Summary and W1–W2. Each component is necessary to tackle CA-FKT: transferring client knowledge to server LoRA adapters while preserving feature alignment, privacy, and avoiding large models on clients.
> >
> > **C2**\
> > See replies to W4 and MC1. All mathematical notation is defined in Section 3.
> >
> > **C3**\
> > See reply to W3. **No client data or logits are ever transferred to the server.** Our approach falls within standard FL privacy assumptions, and since only the classifier head is transmitted, we arguably transfer less information than standard FL techniques. Differential privacy masking can be applied if needed.
> >
> > **C4**\
> > See reply to W8. $\alpha=1$ is already heterogeneous (Hsu et al., 2019). Additional results: top-1 $41.9\%$ ($\alpha=0.5$) / $36.7\%$ ($\alpha=0.1$); top-5 $73.3\%$ / $67.6\%$.
> >
> > **C5**\
> > See reply to W9. EFFEKT stabilizes within rounds 400–500. Additional experiment with $n_r=1000$ attains $42.1\%$ top-1 and $74.9\%$ top-5.
> >
> > **IMPACT**\
> > The Impact Statement section is already included after Section 8 — Conclusions.
> >
> > ---
> >
> > **[R1]** DINOv2 repository. https://github.com/facebookresearch/dinov2.
> >
> > **[R2]** Torchvision - PyTorch, MobileNet v3 model. https://docs.pytorch.org/vision/main/models/generated/torchvision.models.mobilenet\_v3\_small.html
> >
> > **[R3]** Caligiuri et al. FedPromo: Federated lightweight proxy models at the edge bring new domains to foundation models. arXiv:2508.03356, 2025.

---

> > > ### Comment · Reviewer_sLT4 · 2026-07-11
> > >
> > > Thank you to the authors for the detailed replies.
> > >
> > > I appreciate the authors’ explanation of the motivation. However, I think my question is why a CNN proxy on the client is necessary. For example, why not simply use a smaller, pruned, or tiny ViT instead of a model with a different architecture? MobileNetV3-Small requires significantly fewer resources than DINO/ViT, so the paper provides a good example. However, it may not always be possible to find such a lightweight proxy for every FM application. Could the proxy simply be a tiny version of the FM? How is this different from previous work, such as FedGKT for CNNs [1], FedMKT for LLMs [2], or KOALA [3]?
> > >
> > > I also have two points that I would like to further raise.
> > >
> > > - Response to W2: It seems that the authors conflate the paper Meta-FL with federated meta-learning in general. FedMetaNAS [4] addresses client-server architectural mismatch, although it does not provide privacy-preserving knowledge transfer.
> > > Response to W9: Could you provide some intuition for why the proposed method can converge, without necessarily giving a strict formal proof? The anchor point moves in all phases: client-head training, server-side LoRA distillation, and client-server realignment. After the server finishes realignment in each round, the new feature space may drift from the feature space used during client training. Therefore, the solutions obtained in previous rounds of client training may no longer be local minima. Hence, it is not guaranteed that $L^{t+1}<L^{t}$.
> > >
> > > [1] He, Chaoyang, Murali Annavaram, and Salman Avestimehr. "Group knowledge transfer: Federated learning of large cnns at the edge." Advances in neural information processing systems 33 (2020): 14068-14080.
> > > [2] Fan, Tao, Guoqiang Ma, Yan Kang, Hanlin Gu, Yuanfeng Song, Lixin Fan, Kai Chen, and Qiang Yang. "Fedmkt: Federated mutual knowledge transfer for large and small language models." In Proceedings of the 31st International Conference on Computational Linguistics, pp. 243-255. 2025.
> > > [3] Chen, Shaoyuan, Linlin You, Rui Liu, Shuo Yu, and Ahmed M. Abdelmoniem. "Federated knowledge transfer fine-tuning large server model with resource-constrained IoT clients." In International Workshop on Trustworthy Federated Learning, pp. 46-60. Cham: Springer Nature Switzerland, 2024.
> > > [4] Huang, Xinyuan, Jiechao Gao, and Jie Wang. "Federated neural architecture search with model-agnostic meta learning." In 2025 IEEE International Conference on Big Data (BigData), pp. 3452-3461. IEEE, 2025.

---

> > > > ### Author Response · Authors · 2026-07-13
> > > > **Response to sLT4 - Part 1/2**
> > > >
> > > > **Proxy architecture**\
> > > > Concerning the first point, i.e., could the proxy simply be a tiny version of the FM (e.g., TinyViT) rather than a CNN with a different architecture? The short answer is yes. The EFFEKT framework is general and can be applied to any combination of larger and smaller models; it is not architecturally restricted to CNN proxies.
> > > > The only hard requirement is that the client and server feature spaces share the same dimensionality, enabling cross-distillation via a shared classification head. This condition is satisfied by any feature-aligned pair of architectures, including ViT-based proxies. Note that creating these pairs is generally possible thanks to our pretraining step, which handles feature-space alignment. In cases where no sufficiently lightweight architecture exists for a given deployment target, this remains an open challenge shared by all federated approaches requiring on-device training, not only by EFFEKT.
> > > > To confirm this, we ran preliminary experiments replacing MobileNetV3-Small with TinyViT-5m (pretrained on ImageNet-1k, to be as comparable as possible) on the CompCars dataset. EFFEKT works correctly in this setting; however, consistent with the well-known tendency of ViTs to overfit on small datasets, we observed inflated client-side accuracy ($+1.9\%$ on average) at the cost of server-side performance ($-3.4\%$ on average). Notably, the instability of ViTs in this data-scarce regime required us to reduce the pretraining learning rate from the default $5\times10^{-3}$ to $2.5\times10^{-4}$ to obtain stable alignment. Nevertheless, the federated configuration was kept the same, achieving a server-side performance of $39.1\%$ top-1 and $71.3\%$ top-5 - slightly below the CNN-proxy configuration ($43.0\%$ / $75.0\%$), but still substantially above all baselines (best competitor FedPromo: $35.8\%$ / $69.4\%$, Table 1). This confirms that the CNN proxy is a well-motivated practical choice rather than a fundamental limitation of the framework.
> > > > The reason CNN proxies are preferable in our specific setting is that each client dataset contains only a few hundred samples per domain, distributed across 100 clients under Dirichlet concentration $\alpha=1$. In this data-scarce regime, the inductive biases of CNNs act as an effective regularizer, preventing the feature space drift that undermines ViT-based proxies during local training. The server-side FM (DINOv2 ViT-L/14-Reg) is the high-capacity model that benefits from the absence of such biases, generalizing across domains because it was pretrained at scale.
> > > > Note also that CNN models are still very widely used in practice in mobile and resource-constrained environments due to their simplicity.
> > > >
> > > > **Related works**\
> > > > We appreciate the reviewer's effort in drawing connections to related work, and we address each in turn. Regarding FedGKT [R1], while it shares the broad idea of edge devices training small models and a server training a large one via bidirectional KD, EFFEKT differs in three key respects: the server model is a pretrained Foundation Model (DINOv2) rather than a task-specific CNN trained from scratch; knowledge transfer is mediated by LoRA adapters instead of full model updates; and our setting is multi-domain and domain-asynchronous, with clients grouped by domain and LoRAs specialized per domain. FedGKT does not address FM fine-tuning or multi-domain inference.
> > > > Regarding FedMKT [R2], which performs mutual knowledge transfer between a server LLM and client smaller LMs via logit sharing and token alignment, EFFEKT targets computer vision with heterogeneous CNN-FM architectures and, crucially, does not share logits between client and server. Instead, clients upload frozen head weights from which the server computes logits entirely on its own public data, thus achieving a stronger privacy guarantee than logit sharing used in FedMKT.
> > > > Regarding KOALA [R3], which fine-tunes a large server model using small IoT client models as knowledge extractors via reverse KD, EFFEKT extends this direction by using a pretrained FM rather than a task-specific large CNN, adding explicit feature-space realignment through the JA step after each server update, supporting multi-domain LoRA adapters for incremental domain learning without catastrophic forgetting, and providing validation on real edge hardware with measured energy and bandwidth budgets.
> > > > The common thread across these works is large-small model collaboration under resource constraints. EFFEKT's distinguishing contribution lies in combining FM-targeted LoRA updates, cross-architecture feature alignment, and domain-asynchronous multi-domain training in a single coherent framework validated both in simulation and on physical devices.

---

> > > > > ### Author Response · Authors · 2026-07-13
> > > > > **Response to sLT4 - Part 2/2**
> > > > >
> > > > > **W2**\
> > > > > We thank the reviewer for the precise clarification and agree that our earlier response conflated FedMetaNAS [R4] with federated meta-learning in general. We will update our related work discussion to cite it more accurately. That said, FedMetaNAS addresses a different problem: it optimizes a shared architecture across clients via meta-learning and NAS, with knowledge aggregation following FedAvg-style weight averaging. It does not perform privacy-preserving knowledge transfer to a server-side FM, which is the core objective of CA-FKT. Specifically, FedMetaNAS clients train full local models; there is no FM at the server side, and the architectural heterogeneity it addresses is across clients rather than between clients and a server-side FM as in our case.
> > > > > The CA-FKT task therefore remains unaddressed by FedMetaNAS.
> > > > >
> > > > > **W9**\
> > > > > We appreciate the reviewer's request for convergence intuition and provide it here without claiming a formal proof (consistently with FedAvg, MOON, FedHEAL, and FedPromo, none of which provide convergence guarantees in the non-IID setting).
> > > > > The key stabilizing mechanism is the frozen client encoder $E^{d_i}$: since only the classification head $H^{d_i}_{k_j}$ is trained locally (Algorithm A.2), the feature space used during C2S and JA is anchored to the clients' one throughout training. The server-side LoRA update in C2S modifies only 1.57M parameters ($\sim\!\!194\times$ less than the full FM), tightly bounding the magnitude of any single-round drift in the server feature space. The JA step then explicitly re-aligns the aggregated client model $M^{d_i}_k$ and the updated FM $O^{d_i}$ on public pretraining data $\mathcal{D}^{d_i}_p$, restoring feature compatibility before the next round begins. One can therefore think of the system as performing proximal alternating minimization, where the frozen encoder constrains client optimization to a low-dimensional subspace of head parameters, while LoRA constrains server optimization similarly, and JA acts as a soft projection back onto the manifold of compatible feature spaces after each LoRA update. Recall also that C2S and JA are performed in this order, allowing JA to realign the feature spaces of the encoders (anchor for the client training) after any possible drift introduced on the FM by C2S.
> > > > > Moreover, while, as the reviewer notes, the anchor point may move across phases, the scheduling of the learning rate (which decays to 0 at the end of training) provides a practical safeguard against divergence regardless of the convergence of each individual loss component. More specifically, as training progresses, the decaying step size leads to progressively smaller parameter updates in the neighborhood of a local minimum, promoting the stability of the training evolution even as the anchor shifts between rounds.
> > > > > Empirically, Figures A.5–A.6 show monotonic accuracy improvement with narrowing confidence intervals across rounds, and Table A.1 confirms that extending training to $n_r = 1000$ rounds yields only marginal change ($\pm1\%$ vs. 500 rounds), more consistent with convergence to a stable fixed point rather than oscillation. This stability is further verified across a wide range of client architectures, training configurations, and datasets, as reported in Tables A.1–A.4 and Figures A.3–A.8. We will add this intuition explicitly to the paper in a dedicated paragraph following the description of the JA step in Section 4.1.
> > > > >
> > > > > ---
> > > > >
> > > > > **[R1]** He, Chaoyang, Murali Annavaram, and Salman Avestimehr. "Group knowledge transfer: Federated learning of large cnns at the edge." Advances in neural information processing systems 33 (2020): 14068-14080.\
> > > > > **[R2]** Fan, Tao, Guoqiang Ma, Yan Kang, Hanlin Gu, Yuanfeng Song, Lixin Fan, Kai Chen, and Qiang Yang. "Fedmkt: Federated mutual knowledge transfer for large and small language models." In Proceedings of the 31st International Conference on Computational Linguistics, pp. 243-255. 2025.\
> > > > > **[R3]** Chen, Shaoyuan, Linlin You, Rui Liu, Shuo Yu, and Ahmed M. Abdelmoniem. "Federated knowledge transfer fine-tuning large server model with resource-constrained IoT clients." In International Workshop on Trustworthy Federated Learning, pp. 46-60. Cham: Springer Nature Switzerland, 2024.\
> > > > > **[R4]** Huang, Xinyuan, Jiechao Gao, and Jie Wang. "Federated neural architecture search with model-agnostic meta learning." In 2025 IEEE International Conference on Big Data (BigData), pp. 3452-3461. IEEE, 2025.

---

> > > > > > ### Comment · Reviewer_sLT4 · 2026-07-13
> > > > > >
> > > > > > Thank you for the clarification. They make sense to me. Also, please revise the manuscript accordingly regarding theses discussion and preliminary experiments.

---

> > > > > > > ### Author Response · Authors · 2026-07-13
> > > > > > >
> > > > > > > We thank the reviewer for their comments and insight that helped us improve the quality of the manuscript.
> > > > > > > We will update the document (2nd revision) with the new results as soon as the other reviewers either confirm that they are satisfied with the current replies or request additional information to be included in the updated version.

---

### Review · Reviewer_NJsn · 2026-06-08

**Summary Of Contributions:**

This paper proposes EFFEKT, a federated learning framework that adapts a server-side foundation model via lightweight client-side proxy models. The key idea is to avoid deploying the large foundation model on clients. Instead, clients train small proxy models, while the server updates domain-specific LoRA adapters through a two-stage distillation process: Client-to-Server distillation and Joint Alignment distillation.

**Audience:**

Yes

**Audience Explanation:**

Yes. This paper should be of interest to a meaningful subset of the TMLR audience, especially researchers working on federated learning, foundation model adaptation, efficient edge learning, knowledge distillation, and parameter-efficient fine-tuning. The problem itself is timely.

**Broader Impact Concerns:**

NA.

**Claims And Evidence:**

No

**Claims Explanation:**

Mostly yes.

However, some claims need clearer qualification. First, the privacy claim should be narrowed: the paper shows that raw data stay local, but it does not test leakage through uploaded heads or model updates. The authors should either soften the privacy wording or add leakage analysis such as membership inference or model inversion.

Second, the multi-domain evidence is limited to cases with related public pretraining data. The MilitaryAircraft result in Table 2 suggests that when the target domain is far from the pretraining domain, domain recognition and unknown-domain inference can degrade. The authors should discuss this assumption more clearly and test stronger public-target domain mismatch.

Third, the efficiency claim is strong for clients but incomplete for the server. Since C2S and JA add server-side distillation every round, the authors should report server-side cost separately, including wall-clock time, GPU memory, or FLOPs.

Finally, the method appears to assume access to logits/internal representations and trainable LoRA insertion in the foundation model. This should be stated explicitly, since it may not apply to closed-weight API-only foundation models.

**Requested Changes:**

1. clarify privacy claim, report distillation cost compared to baselines, and state foundational model access assumptions (see above)
2. analyze the MilitaryAircraft failure case, which might be due to pretrain public data or downstream task data. Some kind of divergence distribution distance might be helpful
3. Since EFFEKT uses LoRA adapters on the server, it's better to compare against federated LoRA or adapter-tuning baselines under matched compute and communication budgets. This would clarify whether the gains come from the proposed C2S/JA distillation rather than LoRA adaptation alone. Besides, how much performance gain and cost increase do you observer if you replace LoRA with full parameter fine-tuning?

---

> ### Author Response · Authors · 2026-06-09
> **Response to NJsn - Part 1/2**
>
> **Summary:** We appreciate the reviewer's precise and comprehensive summary of our work and hope it will assist the Action Editor in recognizing the significance of our contribution. Following the journal's guidelines, we are currently restricting ourselves to outlining the changes made while we await the completion of all reviews; the revised manuscript with these changes will be uploaded as soon as all reviewes are available.
>
> **Privacy:** We agree that some claims may be overstated; we had to limit the discussion on privacy preservation due to space constraints. Note that we only communicate a subset of the model parameters (no extra information is exchanged over the network), so all privacy guarantees that hold for FedAVG and standard Federated Learning methods also hold for our approach. In fact, our guarantees are arguably stronger, as we only transmit the classifier rather than the entire model.
> We reworded the relevant sections, softening the privacy claims to highlight how privacy is currently guaranteed by:
> (i) the limited number of parameters (theorem 4.1 in Zhang et al. (2024), ArXiv 2407.16735); in our case, the limiting factor of the rank of $\mathbf{J} \in \mathbb{R}^{d \times (B \cdot p)}$ is the number of parameters $d$, as batch size $B$ and input dimensionality $p$ are large;
> (ii) the Local Differential Privacy technique (via Flower's LocalDP), which can also be enabled.
> The privacy guarantor for the latter is based on the moments accountant (composed over $n_r = 500$) with Poisson subsampling amplification ($n_a/n_k = 0.1$) (Abadi et al., *Deep learning with differential privacy*, 2016). The clipping norm is scheduled following the learning rate with max value $N_{c,\max} = 26.9$ for CompCars; the sensitivity is computed as $\Delta_s = \frac{N_c}{65}$ following Wei et al. *Federated learning with differential privacy...* (2020).
> To confirm that the performance degradation is acceptable, we run some additional experiments setting $\delta = 10^{-5}$ and varying $\epsilon$: $\epsilon=50$ degrades performance by $\sim 1\%$, $\epsilon=5$ by $\sim 3\%$, while the more extreme $\epsilon=2.5$ and $\epsilon=1$ yield $\sim 10\%$ and $\sim 20\%$ degradation, respectively.
>
> **Domain Shift & MilitaryAircraft:** We argue our evaluation already covers a wide range of domain shift values (5 heterogeneous domain pairs).
> Nevertheless, we estimated domain shift by computing the CLIP-I and DINO metrics (Ruitz et al., Dreambooth, 2023) between dataset prototypes (i.e., the averages of feature vectors over all samples). This analysis is reported in the revised Appendix.
> More specifically, domain mismatch is heterogeneous, ranging from $0.4$ ($0.84$) [DINO (CLIP-I)] for FGVCAircraft–MilitaryAircraft, to $0.91$ ($0.99$) for CUB200–NABirds. This analysis allows us to infer a reasonable heuristic: a good task-specific pretraining dataset should have DINO similarity $\geq 0.5$ w.r.t. the target.
> Well-aligned domains (StanfordCars–CompCars, Food101–UECFOOD256, CUB200–NABirds, IMNetPets–OxfordPets) hover around $\sim 0.7$ for DINO and $\sim 0.95$ for CLIP-I ; the FGVCAircraft–MilitaryAircraft pair scores $0.4$ and $0.8$, respectively. CLIP-I scores are overall higher since DINO focuses on semantics, while CLIP also encompasses style (here, all images are real and have similar characteristics).
> The MilitaryAircraft degradation has two drivers: (i) semantic misalignment (confirmed by DINO/CLIP-I metrics); (ii) poor sample quality, as the dataset was originally designed for dense tasks (mainly object detection) where the aircraft is often a small object in the background, easily misleading image-level classifiers. Notably, competitor methods also show low performance on this dataset.
> Results from ImageNet pretraining experiments (Table A.6, Appendix) confirm that performance decreases with larger domain shifts.
> We extended the discussion on this dataset in the revised paper.

---

> > ### Author Response · Authors · 2026-06-09
> > **Response to NJsn - Part 2/2**
> >
> > **Server-side Efficiency:** We apologize for the missing details, which have been added to the revised Appendix.
> > Regarding wall-clock time, server-side overhead from C2S and JA scales predictably with distillation frequency (one-every-x rounds $1:x$).
> > Starting from a baseline of $9\text{h}$ (no C2S/JA), adding distillation increases training time to $11.1\text{h}$, $11.7\text{h}$, and $13.2\text{h}$ for $1:10$, $1:5$, and $1:2$ frequency distillation ratios. The proposed EFFEKT configuration ($1:1$) requires $17\text{h}$ ($21.3\text{h}$ with full DINO fine-tuning); ablated variants (EFFEKT no JA, EFFEKT no C2S) require $15.4\text{h}$ and $13.2\text{h}$.
> > Regarding GPU VRAM, the footprint is approximately constant across C2S/JA configurations (classifiers in C2S are negligible vs. the DINO encoder).
> > Meaningful differences arise across optimization strategies: EFFEKT no C2S/JA (i.e., without gradients in the FM) $\sim 11\text{ GB}$, EFFEKT $\sim 14\text{ GB}$ (thanks to LoRA parameter efficiency), full DINO fine-tuning $\sim 19\text{ GB}$ (all values computed using batch size $64$).
> >
> > **FM access:** We expanded the Introduction and Impact Statement section (after Sec. 8 - Conclusions) to explicitly state this limitation: EFFEKT assumes access to logits/internal representations and trainable LoRA insertion in the foundation model, which may not apply to closed-weight API-only foundation models.
> >
> > **Federated LoRA & Full Finetuning:** We target a setting where deploying the server Foundation Model encoder on clients is infeasible due to limited computing capabilities (our proxy encoders have $\sim 100\times$ fewer parameters than the server-side DINO).
> > Since LoRA adapters are architecture-specific and cannot be transferred across different architectures, training them on clients (where the Foundation Model cannot be deployed) is not possible, making federated LoRA approaches inapplicable. Similarly, CNN-targeted variants (e.g., ConvLoRA) would preclude transfer to the server due to architectural mismatch; this is precisely the motivation for our proxy encoder strategy.
> > Furthermore, recall that, unlike our approach, Federated LoRA optimization often encounters issues when applied in conjunction with Differential Privacy, unless specific techniques are used to mitigate the noise composition effect (Sun et al, "Improving LoRA in Privacy-Preserving..." 2024; Wen et al, "Differentially Private Federated Low Rank...", 2025).
> > Regarding full fine-tuning performance, the last row of Table 7 of the main document already reports this comparison. Given the small sample count of target domains and the difficulty of the CA-FKT task, increasing trainable parameters leads to overfitting and degraded performance, achieving slightly lower results than using LoRA adapters.

---

### Review · Reviewer_R5ki · 2026-06-21

**Summary Of Contributions:**

This paper proposes EFFEKT, a federated learning framework where lightweight client-side proxy models are used to update a server-side foundation model through LoRA adapters. The method relies on two distillation steps: client-to-server distillation and joint alignment. Experiments are reported on five fine-grained classification domain pairs, with additional ablations and a small real-device deployment.

The topic is relevant. The setting is also practically motivated: large server models are hard to deploy on clients, and using small proxy models is a reasonable direction. However, I do not think the current paper provides enough evidence for its main claims.

**Audience:**

Yes

**Audience Explanation:**

The problem setting is relevant to a subset of the TMLR audience interested in federated learning, foundation-model adaptation, and edge deployment. The idea of using lightweight proxy models together with server-side adapters is potentially useful. However, interest alone is not sufficient for acceptance here, because the paper’s current evidence does not convincingly support the breadth of the claims.

**Broader Impact Concerns:**

The paper includes an impact statement and acknowledges that model components may leak sensitive information. However, this concern is not sufficiently integrated into the technical evaluation. Since the paper repeatedly emphasizes privacy-preserving learning, the lack of privacy analysis is a notable limitation. I recommend either adding concrete privacy/security evaluation or substantially reducing the privacy-related claims.

**Claims And Evidence:**

No

**Claims Explanation:**

The main issue is that the method depends heavily on task-specific public data at the server. Both C2S and JA use public pretraining data for distillation and alignment. This is not a minor implementation detail; it is central to why the method works. The appendix result with ImageNet pretraining suggests that performance drops substantially when the public data are less aligned with the target domain. This weakens the claim that the method can generally learn new private concepts from clients without strong server-side domain data.

The comparisons are also not fully convincing. The proposed method adds server-side LoRA training and repeated public-data distillation, while most baselines are standard FL optimizers or FedPromo-style aggregation. It is therefore hard to tell whether the gain comes from the proposed bidirectional distillation design, or simply from extra server-side training using relevant public data.

Some reported improvements are strong, especially on CompCars and NABirds. But the picture is not uniformly positive. On OxfordPets, EFFEKT is not the best. On MilitaryAircraft, the Top-5 result is worse than FedAvg+EMA. On UECFOOD256, the Top-5 gain over FedPromo is very small. The paper repeatedly says the improvement is consistent, but the evidence is more mixed.

I am also concerned about the lack of statistical reporting. The main table does not show standard deviations or confidence intervals across the five datasets. A five-seed experiment on one dataset is not enough to establish robustness.

The multi-domain inference result is another weak point. The domain discriminator works well on most domains, but fails badly on MilitaryAircraft when using public-domain prototypes. This is exactly the realistic case the paper wants to support. The target-prototype result is useful as an upper bound, but it does not solve the problem in a private federated setting.

The on-device experiment is a nice addition, but it is limited. It uses 12 devices on one dataset and a controlled LAN setup. This shows the implementation can run on small devices; it does not yet show that the method scales to realistic federated deployments with unreliable clients, network variability, or stronger privacy requirements.

**Requested Changes:**

1. The dependence on task-specific public data is a central assumption of the method. The paper should make this explicit and discuss how realistic this assumption is in the intended federated setting.
2. The ImageNet/public-data-mismatch experiment changes the interpretation of the results. It should be moved into the main paper and expanded to cover weaker or less aligned public-data settings.
3. The current baselines do not fully isolate the contribution of the proposed distillation design. The paper should include simpler server-side LoRA adaptation and public-data distillation baselines.
4. The statistical evidence in the main experiments is not sufficient. Standard deviations or confidence intervals should be reported for all main datasets and key metrics.
5. The assumptions about target-task information are not completely clear. The paper should clarify what the server knows about the target label space, target classes, and domain identity during training and inference.
6. The privacy claims are stronger than what the method demonstrates. The paper should either add privacy leakage analysis or use more cautious language, such as data-local training rather than privacy-preserving training.

---

> ### Author Response · Authors · 2026-06-24
> **Response to R5ki - Part 1/2**
>
> **Summary:**
> We appreciate the reviewer's summary of our work and hope the replies below and the revised document will alleviate doubts on the contributions' evidence.
>
> **Task-Specific Public Data Dependence:**
> We emphasize that the experimental setting is identical to that of the FedPromo reference work, which uses the same evaluation domain pairs.
>
> Regarding generalizability and data quality requirements, we direct the reviewer to Section A.5 of the revised appendix, where we present a detailed analysis of domain-pair similarity based on semantic alignment metrics commonly adopted in Generative AI contexts. Performance does depend on alignment, as correctly noted, but good results do not require a very strong alignment - a DINO score of at least 0.5 is generally sufficient to obtain satisfactory results. We have added an explicit note on this point to the Introduction and Limitations sections.
>
> The considered setup is also realistic: in many practical scenarios, federated learning is employed to recognize particular patterns in specific contexts - it is usually straightforward to obtain a general-purpose dataset of cars, animals, etc., whereas target fine-grained data in specific environments can be distributed across clients. Consider dir example a medical scenario where a public dataset is available server-side, while hospital/patient data must remain local for privacy reasons.
>
> **Baseline Comparisons:**
> All comparisons are conducted in the same setup: federated training considers only the classifier head (attached to the distillation-pretrained proxy encoder), while metrics are computed by attaching the classifier to the server-side foundation model. All competing approaches also undergo distillation-pretraining on task-specific public data, so comparisons are meaningful and fair. FedAVG, FedAVG+EMA, FedProx, MOON, and FedHEAL contextualize standard FL performance; FedPromo serves as an EFFEKT baseline where distributed optimization is performed in the same manner, but no server-side update is applied (which is the core contribution of this work).
>
> Table 5 shows that completely removing distillation leads to a $\sim7\%$ drop in Top-1 accuracy, confirming that its impact is relevant and public data pre-training alone is not sufficient. Note that distillation on task-specific pretraining datasets alone is meaningless, as the class-sets are disjoint from those of the clients - our distillation approaches effectively tackle a data-free Unsupervised Domain Adaptation task on the server (a complementary task to Source-Free UDA [R1]). We refer the reviewer to Tables 5 and A7, and Section A.6 of the updated appendix for further analysis. We have reframed the discussion in Sections 6 and 7 to clarify this point.
>
> **Mixed Results:**
> We agree that in some specific settings, results are less impressive, but on average the improvement is relevant; nevertheless, we have revised the relevant sections to better reflect the actual results.
>
> The MilitaryAircraft dataset should be regarded as a pathological failure case: it was originally developed for object detection, where the plane is often a small background element surrounded by larger foreground objects (people, cars, buildings), making image-level classification very challenging. Furthermore, the semantic mismatch between the task-specific pretraining dataset and the target domain (confirmed in Section A.4) makes it even harder for all approaches; despite this, our method still yields gains over FedPromo in both acc1 and acc5.
>
> Regarding OxfordPets, although our method does not achieve the best overall results, it ranks a close second, with an average improvement of +2.3 points over the techniques in third place (which differ for top-1 and top-5).
>
> **Statistical Evidence:**
> We apologize for this oversight. We originally reported 5-seed results on a single domain due to the stability introduced by the learning rate scheduler, which constrains variance in the final rounds. We have run additional experiments varying seeds across all datasets and report the variance below. Note how it is very low, more precisely below $0.5\%$ in all experiments except one. A new Section A.7 in the updated appendix expands this discussion.
>
> | Dataset | top-1 $\mu$ | top-1 $\sigma$ | top-5 $\mu$ | top-5 $\sigma$ |
> |---|---|---|---|---|
> | CompCars | 42.4 | 0.5 | 74.6 | 0.4 |
> | UECFOOD256 | 62.0 | 0.1 | 88.9 | 0.1 |
> | NABirds | 39.4 | 0.3 | 72.8 | 0.2 |
> | MilitaryAircraft | 12.2 | 0.5 | 33.7 | 0.2 |
> | OxfordPets | 73.8 | 1.5 | 96.9 | 0.3 |
>
> [R1] Yuqi Fang, Pew-Thian Yap, Weili Lin, Hongtu Zhu, and Mingxia Liu. Source-free unsupervised domain adaptation: A survey, 2023.

---

> > ### Author Response · Authors · 2026-06-24
> > **Response to R5ki - Part 2/2**
> >
> > **ImageNet/Data-Mismatch Experiment:**
> > We had to move the discussion on weaker pretraining to the appendix due to space limitations. We have added a compact version of Table A.6 (averaged across datasets) with the relative discussion to Section 7, as requested.
> >
> > **Simpler Server-Side Baselines:**
> > FedPromo can be considered the starting baseline for EFFEKT's dual-distillation strategy. All reported competitors are tested under the same conditions (distillation-pretrained frozen proxy encoders, classifier-only training on clients, performance measured by attaching the classifier to the server-side FM). For a more thorough analysis of the dual-distillation configuration, please refer to Table 5 in the main paper and Table A.7 and Section A.6 of the updated appendix.
> >
> > **Multi-Domain Inference & MilitaryAircraft:**
> > We refer to Section A.5 in the updated supplementary for a detailed discussion on domain similarity. Regarding MilitaryAircraft specifically, the issue is both semantic similarity and scene topology: the dataset was designed for object detection, so the target object is often far in the sky, and classifiers can easily be confused by foreground objects (people, cars, buildings, etc.) that may mislead the domain discriminator toward other domains. Domain similarity with FGVCAircraft is also lower than for the other couples.
> >
> > **On-Device Experiment:**
> > Most competing FL approaches never consider actual real-life deployment, evaluating instead on small simulated toy examples. Our framework implicitly handles client/network reliability: missing client updates are silently discarded, and the client is treated as non-active; distillation strategies work with any $n_a \geq 1$ active clients. If all updates are lost due to network issues, the round is skipped and paused until the server-client link can be re-established. For privacy, we refer the reviewer to the reply to requested change 6 and Section A.4 of the revised appendix.
> >
> > **Target-Task Information:**
> > During training - consistent with standard ML/DL practice - clients must know the total number of classes to initialize their classifier heads, and communicate this count along with their domain ID to the server so it can instantiate an identical (random) classifier for each client before the first round begins. During federated optimization, clients communicate their domain ID for correct LoRA selection (this does not break privacy, as domain categorization is not sensitive). During inference, clients can either share the domain information of the sample with the server (first two columns of Table 2) or choose not to, requiring the server to estimate the domain via our prototypical approach (last two columns of Table 2). We revised the relevant sections to make the setting clearer.
> >
> > **Privacy Claims:**
> > We had to limit the depth of the discussion on privacy preservation due to space constraints. We have expanded the discussion in the appendix (Section A.4). Privacy is guaranteed by:
> > (i) theoretical bounds on the amount of trainable parameters communicated by clients (Theorem 4.1 in [R2]);
> > (ii) the Local Differential Privacy technique (via Flower's LocalDP), which can optionally be enabled and incurs limited accuracy degradation.
> > We have softened the privacy-related claims in the main text to use more cautious language - specifically, using "data-local training" framing where appropriate - while retaining the technical substance.
> >
> > **Broader Impact / Privacy Evaluation:**
> > We refer the reviewer to the reply above and to Section A.4 of the revised appendix, where a more complete treatment is now provided, including the discussion of Local Differential Privacy guarantees and their empirical cost in terms of accuracy.
> >
> > [R2] Xiaojin Zhang and Wei Chen. Theoretical analysis of privacy leakage in trustworthy federated learning: A perspective from linear algebra and optimization theory, 2024. https://arxiv.org/abs/2407.16735